# Role of Mesenchymal Stem/Stromal Cells in Head and Neck Cancer—Regulatory Mechanisms of Tumorigenic and Immune Activity, Chemotherapy Resistance, and Therapeutic Benefits of Stromal Cell-Based Pharmacological Strategies

**DOI:** 10.3390/cells13151270

**Published:** 2024-07-28

**Authors:** Katarzyna Starska-Kowarska

**Affiliations:** 1Department of Physiology, Pathophysiology and Clinical Immunology, Department of Clinical Physiology, Medical University of Lodz, Żeligowskiego 7/9, 90-752 Lodz, Poland; katarzyna.starska@umed.lodz.pl; Tel.: +48-42-2725237; 2Department of Otorhinolaryngology, EnelMed Center Expert, Lodz, Drewnowska 58, 91-001 Lodz, Poland

**Keywords:** anti-cancer therapy, carcinogenesis, cell plasticity, chemoresistance, head and neck cancer (HNC), head and neck squamous cell carcinoma (HNSCC), mesenchymal stromal cells (MSCs), tumour niche, tumour environment

## Abstract

Head and neck cancer (HNC) entails a heterogenous neoplastic disease that arises from the mucosal epithelium of the upper respiratory system and the gastrointestinal tract. It is characterized by high morbidity and mortality, being the eighth most common cancer worldwide. It is believed that the mesenchymal/stem stromal cells (MSCs) present in the tumour milieu play a key role in the modulation of tumour initiation, development and patient outcomes; they also influence the resistance to cisplatin-based chemotherapy, the gold standard for advanced HNC. MSCs are multipotent, heterogeneous and mobile cells. Although no MSC-specific markers exist, they can be recognized based on several others, such as CD73, CD90 and CD105, while lacking the presence of CD45, CD34, CD14 or CD11b, CD79α, or CD19 and HLA-DR antigens; they share phenotypic similarity with stromal cells and their capacity to differentiate into other cell types. In the tumour niche, MSC populations are characterized by cell quiescence, self-renewal capacity, low reactive oxygen species production and the acquisition of epithelial-to-mesenchymal transition properties. They may play a key role in the process of acquiring drug resistance and thus in treatment failure. The present narrative review examines the links between MSCs and HNC, as well as the different mechanisms involved in the development of resistance to current chemo-radiotherapies in HNC. It also examines the possibilities of pharmacological targeting of stemness-related chemoresistance in HNSCC. It describes promising new strategies to optimize chemoradiotherapy, with the potential to personalize patient treatment approaches, and highlights future therapeutic perspectives in HNC.

## 1. Introduction

Head and neck cancer (HNC) comprises a heterogeneous group of malignant neoplasms developing from the mucosa of the oral cavity, pharynx, larynx, nose, salivary glands and oesophagus, with the former three being the most frequent [1,2]. More than 90% of HNCs are squamous cell carcinomas (HNSCCs) that originate from the mucosa of the oral cavity, oropharynx and larynx. All HNSCCs demonstrate morphological and molecular heterogeneity and thus also vary with regard to their clinical course [3]. According to the latest epidemiological data from Global Cancer Statistics (GLOBOCAN 2020), HNSCC is the eighth most common cancer in the world, causing 931,931 new cases and 467,125 deaths in individuals with HNSCC of various origins, accounting for 3% of all cancers and ~1.5% of all cancer deaths. Importantly, analyses clearly indicate that by 2030, the incidence of HNSCC will continue to increase by 30%, i.e., 1.08 million new cases per year (GLOBOCAN; gco.iarc.fr/today (accessed on 30 April 2024)) [4,5]. An increase in the incidence of head and neck cancer (HNC) has been observed in both developed and developing countries worldwide. While HNC has been noted in both women and men, the prevalence is higher in the latter, according to the latest version of GLOBOCAN 2020 and the International Head and Neck Cancer Epidemiology (INHANCE) consortium [4,5]. The epidemiological data regarding the location of HNC clearly show that the highest number of newly diagnosed cases, and the higher cumulative risk of its development, concerns the lip and oral cavity (264,211 new cases annually; the age-standardized rate, ASR = 6.0//100,000 population; cumulative risk for ages 0–74, CR = 0.68% for males). This is followed by cancers of the larynx (160,265 new cases; ASR = 3.6; CR = 0.45%), and then cancers of the nasopharynx (96,371 new cases; ASR = 2.2; CR = 0.24%), oropharynx (79,045 new cases; ASR = 1.8; CR = 0.22%) and hypopharynx (70,254 new cases; ASR = 1.6; CR = 0.19%) region. Regionally, the highest incidence of oral cancer has been recorded in Southeast Asian and Asia-Pacific countries, which is directly related to the chewing of areca nuts (betel quid), with or without tobacco. Unfortunately, the prevalence in these regions will continue to increase with the population size. Similarly, rising rates of HNC were also reported in the USA and Europe, but these were related to an increase in the incidence of oropharyngeal cancer associated with papillomavirus (HPV) infection. Interestingly, epidemiological studies clearly indicate a growing trend in the incidence of HPV-dependent HNC tumours in the last decade and that this has been accompanied by a simultaneous decrease in the incidence of HPV-independent tumours, especially in countries such as the USA, Canada, Hong Kong and Korea. Epidemiologists also indicate that in the next twenty years, the majority of HNC cases will be etiologically related to HPV infection and that European countries, such as the UK, will be particularly badly affected, with the incidence of oropharyngeal cancer exceeding the number of diagnosed oral cancers. It should also be noted that the number of laryngeal cancer cases has increased by 23% worldwide in the last ten years. However, when the rates are adjusted for age, the number of new cases of laryngeal cancer is decreasing, especially in countries with higher sociodemographic standards and more health-promoting behaviours related to limiting smoking and alcohol consumption. Additionally, it is well known that the risk of developing HNC increases with age in various populations, with most cases diagnosed in older age groups. However, the peak incidence of oropharyngeal cancer occurs about ten years earlier, around the age of 60–65 [4,5].

Unfortunately, despite great advances in surgical procedures and application of innovative therapeutic modalities of chemoradiation, this common type of malignancy is associated with high mortality and morbidity due to the common nodal metastases and local neoplastic recurrences and the development of chemoradioresistance. Importantly, approximately 60–70% of individuals are diagnosed at an advanced stage of neoplastic disease, i.e., WHO classification stages III and IV; as such, the condition is characterised by low global overall survival, with five-year tumour-free survival rates not exceeding 40–60%, with the hypopharynx experiencing the worst outcomes. Although there are clear differences in the HNC mortality rates, especially in European countries, the combined sex-related HNC mortality has fallen by 11% overall. At the same time, it is worth noting that the last decade has seen a gradual increase in mortality rates compared to the period 2006–2018. For instance, in the UK, HNC mortality rates have tended to increase over the last decade, reflecting an increasing incidence and static survival rates. However, the death rates per 100,000 persons vary with regard to the location of the HNC: the highest rates were recorded for cancer of the lip and oral cavity (125,022 cases annually; ASR = 2.8/100,000 population; cumulative risk for ages 0–74, CR = 0.32%) for men. The cumulative risk of dying from cancer was also high for laryngeal cancers (85,351 cases; ASR = 1.9; CR = 0.23%), followed by the nasopharynx (58,094 cases; ASR = 1.3; CR = 0.16%), the oropharynx (39,590 cases; ASR = 0.9; CR = 0.11%), and hypopharynx (32,303 cases; ASR = 0.7; CR = 0.09%) [4,5,6,7,8,9].

HNSCC is most commonly observed in tobacco and alcohol users, and in individuals exposed to environmental contaminants and other carcinogen-containing chemicals; these have usually accounted for 90% of cases of HNSCC, in accordance with the National Comprehensive Cancer Network (NCCN), and are usually associated with a poor prognosis [10,11,12,13]. Importantly, research clearly shows that tar and tobacco smoke have in their composition over 7000 toxic substances, 60% of which are important carcinogens. Tobacco onco-toxins such as benzo(a)pyrene, a crucial polycyclic aromatic hydrocarbon (PAH), as well as tobacco-specific nitrosamines (TNA) and N′-nitrosonornicotine (NNN), considerably up-regulate the risk of tumour development and further neoplastic growth and the occurrence of nodal and distant metastases by inducing a phenotype similar to epithelial-mesenchymal transition (EMT) [14,15,16]. Alcohol causes atrophic epithelial changes, acting as a co-carcinogen that increases the risk of HNSCC in concomitant smokers. Interestingly, the main metabolite of ethanol, acetaldehyde, is also a strong mutagen. The interaction of these two chemicals increases HNSCC development by up to 35 times [14,15,16]. According to data provided by the Cancer Genome Atlas Network (TCGA), tobacco-related cancers show aberrations in a number of genes, including cell cycle regulators (CDKN2A and CCND1), regulators of cell division and viability (TP53, HRAS, PIK3CA and EGFR), genes related to cell differentiation (NOTCH1) and Wnt signalling pathway genes [3]. They are also associated with loss of chromosome 9p, resulting in a reduction of p16 (CDKN2A) expression, and duplication of chromosome 7p, favouring increased activity of the epidermal growth factor receptor (EGFR). In addition, tumours are commonly associated with mutations of proto-oncogenes, i.e., c-MYC, c-KIT, HER-2, RAS, BCL-2, and STAT3, and the inhibition of anti-oncogenes, i.e., tumour suppressors RB1, P53, INK4, PTEN and CDKN2A, as well as others promoting the pro-inflammatory neoplastic milieu [3,17,18].

In recent years, human papillomavirus (HPV) infection, primarily HPV-16 but also HPV-18, has been observed in the epithelial of oropharyngeal location (OPSCC). It is more common in middle-aged patients, demonstrating different development and therapeutic effects, being much more commonly diagnosed in the earlier stages, and it is characterised by better prognosis due to the greater sensitivity to chemoradiation and immune checkpoint (ICI) blockade [19,20]. This neoplasm is histologically and clinically exceptional and accounts for as many as 38–80% of newly diagnosed HNSCC cases [19,21]. OPSCCs contain the HPV genome, which contributes to tumour metaplasia through the activity of viral oncoproteins E6 and E7 [22,23]. OPSCC activity is associated with the integration of HPV genomic DNA with that of host epithelial cells. Amplified activity of HPV16/18 virus antigens E5, E6 and E7, which constitute oncoproteins, induces malignant transformation. HPV16/18 E6 and pRB1 degrade proteins that regulate the cell cycle, e.g., the tumour suppressor p53, through HPV16/18 E7; this inactivates crucial intracellular signalling pathways associated with cell cycle control [15,24,25,26]. In addition, the HPV16/18 E6 protein disturbs the activity of the c-MYC oncogene, increasing the transcription of the human telomerase catalytic subunit (hTERT), which contributes to the immortalization of neoplastic cells and disrupts the activity of CDK, cyclins and E2F transcription factors. It has also been found that MYC reverses the inhibitory function of CDK p27^KIP1^ and p21^CIP1/WAF1^ [25,27]. According to the TCGA, the characteristic gene aberrations for HPV-associated cancers include PIK3CA, DDX3X, CYLD and FGFR mutations [28]. HPV-positive neoplasms also express amplification of chromosome 3q and the loss of chromosomes 16p, 16q, 14q, 13q and 11q [17,18].

In the earlier clinical stages (I and II), the therapeutic treatment procedure for HNSCC cases is mostly surgical, being commonly based on modern low-invasive surgical techniques. However, at higher clinical stages, the treatment is more likely to include radiotherapy (RT) or concurrent cisplatin-based chemotherapy with surgery, especially in patients whose tumours are locoregionally advanced or have positive margins. Unfortunately, as advanced HNSCC frequently demonstrates resistance to conventional treatment, i.e., platinum (CDDP)-based chemotherapy plus 5-fuorouracil (5-FU), extensive research is still being carried out to develop new molecularly targeted therapies. So far, Cetuximab, a monoclonal antibody against the epidermal growth factor receptor (EGFR), and the PD-1 inhibitors Nivolumab and Pembrolizumab have been approved for the treatment of clinically advanced HNSCC. Unfortunately, only Cetuximab shows limited effectiveness in patients with unresectable and recurrent HNSCC [9]. Importantly, taking into account the biological nature of HPV-positive cancers and the association with the prognostic parameters of HPV-related OPSCC, the traditional staging of HNSCC using the tumour-node-metastasis classification has been supplemented by the new 2017 AJCC/UICC staging system: this eighth edition of the American Joint Committee on Cancer (AJCC) guidelines recommends a de-intensified treatment protocol for patients with HPV-related OPSCC to reduce the long-term associated morbidity [29,30,31]. However, recent clinical trials (RTOG 1016 and De-ESCALaTE) indicate that individuals with HPV-positive OPSCC present a more complex therapeutic problem, as a large cohort of patients who received de-escalation de-intensified therapy had significantly worse results than those who received standard care [32,33]. Moreover, patients with HPV-induced advanced OPSCC and a second primary cancer in any head and neck region demonstrated a lower pT/pN stage compared to those with a single primary tumour [34].

Regardless of its origin and type, the HNSCC tumour microenvironment (TME) consists of various stromal cells that may regulate neoplastic development, progression and invasiveness. Interestingly, recent studies clearly indicate that the presence of microenvironmental stemness cells in the tumour niche, and their potential to modulate carcinogenesis, tumour advancement and aggressiveness, may also influence the response to cisplatin-based chemotherapy and accompanying radiotherapy (RT), as well as to new molecular targeted procedures. One component of the neoplastic ecosystem comprises the tumour-associated mesenchymal stromal cells (MSCs): a small but critical subpopulation of the extracellular matrix (ECM) cells capable of self-renewal, multilineage differentiation and regeneration. The growing field of research on their role in relation to tumorigenesis in HNSCC has resulted in a few new and interesting studies over the last decade. Unfortunately, the role of the MSC in the tumour milieu in HNSCC pathogenesis, therapeutic resistance and patient prognosis remains unclear. Even so, the activities of stemness ECM cells present in the tumour microenvironment may influence future therapeutic strategies for HNC. Nevertheless, MSCs are also reported to exhibit dual roles as both pro- and anti-cancer factors in human cancers, including HNSCC [35,36,37,38,39].

This narrative review aims to present a comprehensive picture of the latest relevant literature regarding the importance of tumour-associated mesenchymal stromal cells (MSCs) in the initiation of carcinogenesis, determination of more stemness and invasive phenotype and the activation of regulatory mechanisms involved in the stemness-related development of resistance; the review also covers the chemoresistant effects to cisplatin-based chemotherapeutics contributing to treatment failure and the potential therapy of HNC of various site origins. The corpus of research comprises a wide range of recent pre- and clinical early trials, experimental and molecular studies, including in vitro cell culture models and animal or in vivo models of HNSCC. The work serves as a compendium of knowledge on the etiopathogenesis of HNC related to the activity of MSCs, based on the latest publications and available data on the topic, selected by the author of the article. The following inclusion criteria were applied: (a) written in the English language; (b) studies based on humans or animal tissues (both in vivo and in vitro); (c) cohort studies; (d) retrospective studies; (e) prospective studies; and (f) patients with a confirmed diagnosis of HNC. This review also presents up-to-date knowledge and focuses on the possibilities of pharmacological targeting of stemness-related chemoradioresistance in HNC. The final search (conducted on 30 April 2024) included the most valuable and highest-rated peer-reviewed articles published in the last decade (January 2014 to April 2024), all of which are accessible via the PubMed/Medline/EMBASE/Cochrane Library databases. The databases were searched using the following keywords: “head and neck neoplasm or HNSCC”, “squamous cell carcinoma of head and neck”, “head and neck cancer”, “head and neck carcinoma”, “HPV-related tumours”, “mesenchymal stromal cells”, “MSC”, “stem cell resistance”, “therapeutic or drug resistance”, “stemness potential”, “stemness markers”, “head neck cancer stem cell resistance”, and “HNC stem cell resistance”. We identified additional records by cross-references. Tumours of the oesophagus and nasopharyngeal region, such as squamous cell carcinomas, were included in the article due to their similar histology to the group of the most common head and neck squamous cell carcinomas (HNSCCs) and for comparison of the research results on MHCs. The following exclusion criteria were applied: (a) unpublished articles or conference proceedings; (b) case reports, case records, and letters to the editor; (c) studies where the patient’s diagnosis was uncertain (e.g., no histopathological confirmation); and (e) abstracts. The papers that merely reported an association between HNC carcinogenesis, progression and resistance phenotype and the activity of mesenchymal stromal/stem cells were also excluded, and the work focused on papers that reported evidence generated with the use of stem cell subpopulations. This work discusses the identified issues in detail in order of increasing clinical credibility.

## 2. Characteristics of Mesenchymal Stromal Cells (MSCs)

### 2.1. The Identification, Distribution and Pathophysiological Role of Mesenchymal Stromal Cells (MSCs)

Mesenchymal stromal cells (MSCs) constitute a small population of multipotent, mobile cells, but an interesting one that has been intensively studied. MSCs participate in tissue homeostasis regulation, tissue/organ repair, maintenance of blood vessel integrity and immune system modulation [40,41,42,43]. The latest observations indicate that inflammatory signals at the lesion site promote the chemotactic motility and migration of MSCs to the site of damage and cancer niche, where the production and secretion of signalling molecules occurs, as does subsequent differentiation into various cell types. The relocation of the MSCs to the damaged or inflamed site is controlled by stimulatory signals, which also influence the host response and promote naïve MSC differentiation. The chronic inflammation accompanying the tumour microenvironment is a key condition favouring the colonisation of the TME by MSCs; indeed, numerous studies indicate that tumour-associated mesenchymal stromal cells can be recovered from various neoplastic samples. The main sources of MSCs are bone marrow, adipose tissue and dental pulp, but they can also be recovered from other sites, i.e., muscular, bone, and cartilage tissue in vivo; however, the most commonly used tissue sources for obtaining MSCs are BM-derived MSCs (BM-MSCs) and adipose tissue-derived MSCs (A-MSCs) [38,40,44,45,46,47,48]. It is postulated that MSCs are most often located near blood vessels, in permeating arterioles and within the tunica adventitia of arteries. Mesenchymal cells from these tissues have the capacity to divide symmetrically (forming two stem cells or two differentiated cells) or asymmetrically (forming one stem cell and one differentiated cell). They are also believed to be able to transform into certain tissue lines and have motility properties [41,46,48].

Since the ability of MSCs to transdifferentiate cells is controversial, the International Society for Cellular Therapy (ISCT) has suggested that the terminology for these cells should only refer to cells meeting specific stemness criteria [49,50,51]. The most important of these are the three minimum criteria: (1) MSCs must demonstrate plastic-adherence when grown in vitro; (2) MSCs must express surface antigens such as CD73, CD90 and CD105, while lacking CD45, CD34, CD14 or CD11b, CD79α, or CD19 and HLA-DR antigens; and (3) when cultured under well-defined conditions, MSCs must be able to differentiate into specific mesodermal cell types after stimulation, i.e., adipocytes, chondrocytes, and osteoblasts. In addition to the mesodermal lineage, MSCs have high plasticity and are capable of transforming into cells of non-mesodermal origin, i.e., ectodermal and endodermal lineages, such as neuronal cells, cardiomyocytes, hepatocytes, or epithelial cells. Further in vitro research into MSCs has allowed their identification based on other features, like the presence of a subset of other characteristic surface markers and the ability to form colonies [40,48,52,53,54]. One set of antigens suggested for MSC recognition includes the superficial proteins, for instance, smooth muscle actin (SMA), Gremlin-1 (GREM1), Meflin (ISLR), PDPN (Podoplanin), STRO-1, and stage-specific embryonic antigen 4 (SSEA-4) [55,56,57,58,59,60,61]. Interestingly, the presence of a key SMA marker allows MSCs to be differentiated from activated fibroblasts/myofibroblasts in the ECM. Moreover, SMA up-regulation promotes the conversion of mesenchymal stromal cells into myofibroblasts and cancer-activated fibroblasts (CAFs) [62,63,64,65]. However, MSC identification can be hampered by the fact that they present similar surface markers and microvascular pericytes to fibroblasts [66,67,68]. Even so, fibroblasts have been differentiated from MSCs by co-expression of fibroblast activation protein alpha (FAP-α) and fibroblast specific protein 1 (FSP, also known as S100A4) [69,70]. Gremlin-1 (GREM1) is also indicated to be an important candidate marker of MSCs [56,57]. Unfortunately, markers such as ISLR, PDPN, STRO-1, and SSEA-4 are also present in other cell types and hence have limited value.

### 2.2. Dual Roles and Bidirectional Effect of Mesenchymal Stromal Cells in the Tumour Microenvironment

Tumours, including solid tumours in the head and neck region, are associated with a complex microenvironment, including heterogeneous cell populations, such as neoplastic cells, transformed stromal cells/cancer stem cells, immunocompetent cells, blood vessels and lymphatic vessels. Highly specialized communication and modulation can be found between cells present in the tumour ECM, which is maintained through secreted cytokines, chemokines and various other mediators by paracrine signalling or through cell–cell interaction [38,47,48]. These paracrine agents can be directly secreted into the tumour milieus or released via extracellular vesicles (MSC-EVs) [52]. Recent studies have shown that after transformation in the neoplastic niche, the activity of naïve chemoattractant-induced MSCs induces production of various pro-/anti-inflammatory cytokines, chemokines or mediators that provide the plasticity, variability and uniqueness of the tumour milieu. These modulatory cells may enhance resistance to cisplatin-based chemotherapy, thus favouring cancer progression and therapeutic failure; the activation of pro-tumour mechanisms is a common cause of shortened survival.

Unfortunately, the primary role of MSCs in the initiation of carcinogenesis and subsequent stages of tumour development has not been fully elucidated; despite this, a few studies indicate that MSCs have a positive effect on the inhibition of neoplastic lesions by activating anti-cancer mechanisms [71,72,73,74]. Indeed, numerous recent studies indicate that cancer-associated MSCs appear to have dual effects on tumour progression in various in vitro and animal or in vivo neoplastic models, and the pleiotropic consequences of MSCs may therefore confer pro- or anti-tumour functions to cells in the tumour microenvironment [38,52,75,76]. The biomolecules produced by tumour-associated MSCs and cancer cells are often named as, in accordance with the nomenclature from the English literature, an active “secretome”; such a mixture includes key immune-modulating agents, pro-/antiangiogenic factors, pro-survival biological agents or soluble factors inhibiting mobility and cancer cell survival and extracellular matrix modulators [77,78,79,80]. Crosstalk between tumour cells and mesenchymal MSCs, which can alter the behaviour of TME cells and regulate tumorigenesis through the secretion of secretome biomolecules, has also been observed in tumours of the head and neck region. Examples of soluble agents and modulators in HNC include IL-6, IL-8, plated-derived growth factor (PDGF), beta 2 microglobulin (B2M), cellular communication network factor 2 (CCN2), fibroblast growth factor 19 (FGF19), stromal cell growth factor-beta (SCGF), transforming growth factor-beta 1 (TGF-β1), stromal cell-derived factor 1 (SDF-1) and matrix metalloproteinases (MT-MMPs); these have been found to directly regulate the epithelial–mesenchymal plasticity, proliferation, invasion and migration and even drug resistance in HNSCC cells [81]. To summarize, the composition of pro- or anti-tumorigenic mediators secreted in the secretome, and their action, may promote or inhibit carcinogenesis in the neoplastic niche occupied by tumour-associated MSCs. Moreover, these components of the secretome may constitute an important target for modern anti-cancer drugs, which will be discussed in detail in the following paragraphs.

#### 2.2.1. Pro-Tumour Activity of MSCs

Like damaged or injured tissues, tumour cells have a chemoattractant effect on mesenchymal tumour stromal cells, activating their recruitment to the neoplastic niche. One of the most characteristic signalling pathways causing their mobilisation to the tumour microenvironment involves the participation of cytokines, chemokines or chemokine receptors, i.e., CXCL motif chemokine ligand 1/2/12 (CXCL1/2/12), chemokine receptor 4 (CXCR4), and chemokine ligand 1/5/7/8 (CCL1/5/7/8). More importantly, MSCs may act directly via cell-to-cell contact or release numerous soluble factors with pro- and anti-tumour effects into the extracellular matrix; these influence survival, proliferation and angiogenesis, as well as metastasis, and they can stimulate and control other cellular functions, thus increasing tumour aggressiveness. These paracrine mediators may be secreted directly into the tumour niche or secreted via extracellular vesicles (MSC-EVs), i.e., exosomes, microvesicles, and apoptotic bodies: these are cell-derived membrane-surrounded balls that deliver bioactive molecules to recipient cells [52,82]. A few factors also stimulate MSCs to migrate towards tumour tissue and may inhibit apoptosis in cancer cells. These biomolecules control the chemotaxis and activity of immune and environmental cells in the TME by promoting the production of factors that lead to immunosuppressive effects and tumour cell survival. These include growth factors such as transforming growth factor beta 1 (TGF-β1), platelet-derived growth factor (PDGF), epidermal growth factor (EGF), basic fibroblast growth factor (bFGF) or vascular endothelial growth factor (VEGF), together with extracellular matrix molecules such as such as matrix metalloproteinase 2/3/7/9 (MMP-2, MMP-3, MMP-7, MMP-9) [52,83]. Numerous studies have also shown that MSCs isolated from squamous cell carcinomas can mediate cancer progression by secreting pro-inflammatory and pro-angiogenic cytokines such as IL-6 and IL8, which also promote recruitment of leukocytes, i.e., tumour-associated macrophages (TAMs), cancer-associated fibroblasts (CAFs), T cells and neutrophils: the cells that facilitate tumour initiation and progression. Moreover, it was confirmed that MSCs recruited from human cancers can actively increase the expression of immunosuppressive agents, i.e., IL-10, TGF-β1 and indoleamine 2,3-dioxygenase (IDO), and angiogenic factors, i.e., TGF-β1, VEGF, angiopoetin-1, endothelin-1, and IL-6. These facilitate further tumour progression and growth through enhanced angiogenesis. They also encourage the formation of further vascularization and the local/regional and general spread of tumour cells [84,85,86]. Furthermore, the development and invasiveness of neoplastic cells may also be connected with a wide spectrum of cancer-related signalling pathways involved in the crosstalk between MSC and tumour cells, such as phosphatidylinositol 3-kinase/protein kinase B/mammalian target of Rapamycin (PI3K/AKT/mTOR), the Janus kinase/signal transducers and activators of transcription (JAK/STAT), Wnt/β-catenin signalling, the Hippo pathway, and MYC and NF-κβ signalling cascades [87]. PI3K/AKT pathway activity appears to be associated with the acquisition of tumorigenic properties by tumours, such as increased rates of cell proliferation, drug resistance, and stem cell-like phenotypes [88]. In one in vitro analysis, it was reported that the cell culture conditioned medium from bone marrow-derived MSC (BM-MSC-CM), promoted the invasiveness of HNC by activating the PI3K/AKT signalling pathway [89]. Additionally, MSC-CM co-cultured with HNC tumour cells increased the invasiveness by enhancing cell proliferation, migration, and epithelial–mesenchymal transformation, and by altering the expression of cell cycle regulatory proteins, as well as by inhibiting apoptosis. Furthermore, these phenomena were induced through the activation of the PI3K/AKT/mTOR pathway, and they were observed to enhance the expression of periostin (POSTN) and N-cadherin in the EMT around the tumour tissues [89]. Interestingly, MSCs can also promote increased tumour aggressiveness and the formation of metastases from cancer cells when engulfed by the neoplastic cells. In such cases, this assumption of the MSC can alter the transcriptome profile of cancer cells, particularly those related to oncogenic pathways, and enhance the epithelial-to-mesenchymal transition, stemness, invasion and metastasis. A recent study found MSC-engulfing tumour cells to be characterised by certain up-regulated genes encoding cell surface and extracellular proteins: MSR1 (CD204), WNT5A, ELMO1, IL1RL2 (IL-36), ZPLD1, and SIRPB1 (CD172); in addition, high activity of MSR1 and WNT5A was found to be significantly related to worse metastasis-free survival in cancer patients [90].

#### 2.2.2. Anti-Tumour Activity of MSCs

On the other hand, several recent publications have concluded that tumour-associated MSCs may also have inhibitory effects on the tumorigenic cell phenotype. These may be realised through their cytotoxic effects on cancer cells, thus inhibiting the initiation, development and progression of cancer. This can also provide a starting point for new strategies for human cancer treatment [52,72,87]. Compelling evidence shows that MSCs utilise various anti-tumour mechanisms to induce apoptosis, repress tumour growth, inhibit angiogenesis and suppress neoplastic cell proliferation. Interestingly, the most important signalling factors found to repress tumour progression and aggressiveness include MSC-related paracrine-soluble secreted factors and MSCs-derived molecules released from exosomes; these include proven anti-proliferative factors such as Dickkopf-related protein 1 (Dkk-1), a soluble Wnt antagonist, phosphatidylinositol 3,4,5-trisphosphate 3-phosphatase (PTEN), and bone morphogenetic protein (BMP), as well as cytotoxic agents such as TNF-related apoptosis-inducing ligand (TRAIL) and TNF-α. In addition, anti-angiogenic factors or immunomodulatory agents have been recorded, as well as some known to inhibit cancer-related signalling, such as PI3K/AKT, Wnt/β-catenin and the JAK/STAT pathway [52,72,76]. MSCs can also express anti-tumour activity by inhibiting tumour angiogenesis via down-regulation of the PDGF/PDGFR axis, thus restricting vascular growth. Studies have shown that MSC treatment reduced the PDGF-BB protein concentrations in tumour lysates, with the levels correlating with reduced activation of PDGFR-β and the isoform of the target protein AKT [91]. Also, the multifunctional cytokine TGF-β1, released from MSCs, can play a dual role in human tumorigenesis. Indeed, studies have shown that the type III TGF-β receptor (TGFBR3) and its shared extracellular domain (sTGFBR3) may maintain epithelial homeostasis by regulating the TGF-β pathway. Restoring TβRIII expression in human cancer cells inhibited the tumour invasiveness and the growth of blood vessels from the existing cancer vasculature in vitro, as well as metastasis in vivo, thus inhibiting the development of an immunotolerant tumour microenvironment [92]. Hence, due to the controversial roles of MSCs in human cancers, future extensive research should focus on the key molecular mechanisms regulating MSC-mediated interactions between mesenchymal stemness cells and tumour cells.

It should be emphasised that a number of experimental factors can influence discrepancies in the observations regarding the ability of MSCs to promote or inhibit neoplastic disease growth. These may include inter alia differences in the choice of experimental tumour in vitro and the animal or in vivo models, the origin of the cancer or HNSCC cell lines, the varied source of MSC tissue, the dose or duration of MSC therapy, the method of obtaining cells, the control group selected, or an insufficient number of studied groups [93,94]. MSC-mediated signalling changes might also play dual roles in obtaining a pro- or anti-cancer cell phenotype; this discrepancy might be attributable to differences in cancer types from the same region and the activation status of key regulators of pathway cascades. The origin of the tumours in the head and neck region is also an important consideration when comparing systems, even within a single squamous type of HNC, as this can have a considerable influence on their cell biology and molecular activity. Moreover, cancer patients are generally subject to a more complex local situation in the tumour microenvironment, and to a peripheral one in the bloodstream, and as such, other factors, e.g., local hypoxia and the supply of oxygen and nutrients to tumour cells, must also be taken into account. Such variation may also be the reason for the failure of proposed modern MSC-based therapies, which may induce unforeseen and unknown intracellular and molecular changes in MSCs and tumour cells [95,96,97]. Therefore, researchers into individualised treatment approaches must exercise particular caution when drawing conclusions regarding the therapeutic effects of MSCs on cancer, and they should recognise that MSC-based therapies still remain a challenge.

A schema of the dual roles of MSCs in the TME and their pro- and anti-tumorigenic activity is shown in Figure 1A.

### 2.3. Modulation of Immune and Inflammatory Cells by Mesenchymal Stromal Cells

Lots of in vitro studies show that tumour-associated mesenchymal stromal cells may affect the immune response in dendritic cells (DCs), tumour-associated macrophages (TAMs), active CD65^dim^/CD16^bright^ NK cells, activated T CD4^+^/CD8^+^(CTL) and B lymphocytes via their secretome; they also confirm that both direct action via cell-to-cell contact or secretion of numerous soluble factors and MSC-derived exosomes into the extracellular matrix can control the activation, proliferation, differentiation and function of both the immune cells themselves and the surrounding cells at the lesion site [98,99,100,101,102]. Interestingly, the biologically active agents and mediators produced and released by the MSCs play a dual role in inhibiting the host defence by interfering with the adaptive and innate immune response [98,103].

Numerous associated immune agents with an innate and adaptive immune response are regulated through MSC-derived molecules; these include IDO, NO (nitric oxide), prostaglandin PGE2, TGF-β1, cytokines IL-10 and IL-6, human leukocyte antigen (HLA-G), iNOS, IL-β1, HO1 as well as other immunomodulators or immunosuppressive agents. Most of these are membrane proteins or factors produced during the immune reaction to infection and autoimmunity, or during tumorigenesis; these have been found to proactively regulate the proliferation and action effects of various immune cell subpopulations via the production of modulatory factors determining angiogenesis, cell death (apoptosis) and tissue regeneration in defined medium components and a heterogeneous culture environment in vivo and in vitro [98,99,100,101,102,103,104]. It was noted that they may also release trophic factors that can increase cell survival, cell proliferation and tissue fibrosis. Furthermore, MSCs can also directly inhibit immune cell activity through cell-to-cell interaction. This direct communication between MSCs and T cells can often inhibit immune cell proliferation: the interaction between programmed death-1 (PD-1) molecules and PD-L1 and PD-L2 ligands can induce apoptosis in effector CD4^+^ and CD8^+^ T lymphocytes. Moreover, various types of T cell anergy can be provoked by MSCs; in such cases, the cells inhibit the expression of CD86 and CD80 in antigen-presenting cells (APCs), i.e., DCs and TAMs [98,99,104].

#### 2.3.1. The Adaptive Immune Response

A characteristic feature of tumour-associated MSCs is their strong immunosuppressive and immunomodulatory properties, which allow cancer cells to escape immune surveillance. MSCs can be stimulated within the tumour niche by pro-inflammatory cytokines such as TNF-α, IFN-γ or IL-1β, which are produced by both macrophages and tumour cells. The activity of MSCs in the tumour milieu is driven by immunological stimuli, e.g., IFN-γ and lipopolysaccharide (LPS); their activation was also observed to increase under oxidative, heat shock, hypoxic and nutrient-deprived conditions, which can occur in solid tumours. This was associated with elevated activity of cytoprotective genes and increases in MSC potency caused by greater secretion of compensating factors [98,99,100,101,105]. It is well known that the active mesenchymal stromal cells in the tumour milieu inhibit the adaptive immune response by secreting mediators contained in exosomes (MSCs-EV) and soluble factors, such as IDO, TGFβ1, TNF-α, IFN-γ, PGE2, NO, HLA-G, HGF, IL-1β, IL-1α, IL-4 and IL-6; they can also do so by interacting with various immune cell types, including T cells, B cells, DC cells, NK cells, monocytes and TAMs. This MSC activity constrains dendritic cell maturation, reduces T cell proliferation, enhances macrophage activation and polarisation of M1 towards M2 and facilitates neutrophil mobility. It also affects the regulation of CD56^dim^/16^bright^ NK cells and invariant natural killer T (iNKT) cells, and shifts the balance of T cell differentiation from pro-inflammatory Th1 to anti-inflammatory Th2. Furthermore, it enhances the maturation of T helper cells into the CD4^+^CD25^+^Foxp3^+^Treg pathways, which can inhibit effector T cell responses and thus reduce anti-tumour immunity [98,99,100,101,102,103,104]. In vitro studies have shown that this shift can also regulate the proliferation of B cells and their differentiation into mature plasma cells, and it can inhibit IgG immunoglobulin secretion from plasma cells, leading to the development of a suppressive phenotype by cell cycle arrest [106]. In addition to maintaining chronic inflammation, it was also shown that MSCs inhibited the effector activity of cytotoxic T cells (CTLs) in mouse models by increasing the secretion of MSC-derived soluble factors, such as arginase 1 (ARG-1) and inducible nitric oxide synthase 2 (iNOS2) [107,108,109]. MSCs also secreted CXCL3, a member of the growth-related oncogenes (GRO-γ), constraining the differentiation and role of MSCs, facilitating their shift to a myeloid-derived suppressor cell phenotype (MDDCs) [110]. MSCs can also induce the recruitment of inhibitory immune cells (MDSCs) through CCL2 signalling, which inhibits the anti-tumour function of T cells [111]. This was accompanied by an increase in the expression of genes related to MSCs, including COX2, IDO-1, programmed death ligand (PD-L) 1 and 2, and matrix metalloproteinase 9 (MMP-9) in human MDDCs. MSC-derived hepatocyte growth factor (HGF) also determined the reactivity of these cells by directly binding the HGF receptor and c-met, and by causing an increase in the STAT3 phosphorylation in MDSCs [108,109]. It has also been found that the soluble PD-1 ligands produced by MSCs effectively inhibit the effect of IL-2 on T cell activation. Soluble PD-L1 and PD-L2 produced by MSCs blocked the parallel stimulation of AKT signalling pathways, which also inhibited T cell activation and proliferation. While PD-1 and PD-L1 have also been found to down-regulate T cell activity, it has recently been found that the immunosuppressive effect of MSCs is partially dependent on cytotoxic T cell antigen 4 (CTLA-4). CTLA-4 occurs on the surface of T lymphocytes activated by contact with an antigen and inhibits the further response of the lymphocyte [112,113]. This expansion process also appears to be supported by IL-6, a cytokine constitutively secreted by MSCs, which was found to inhibit the apoptosis of neutrophils and lymphocytes [98,99,100]. Also, MSC-derived IL-6 also activates neutrophils through the STAT3-ERK1/2 signal transduction pathway, and it shifts their immunosuppressive polarization towards tumour facilitation and supporting cancer progression [114]. Also, NO synthesis and secretion is driven by inducible NO synthase (iNOS) following secretion of various inflammatory molecules, such as IFN-γ, IL-1 and TNF-α, which subsequently suppress T cell function [98,115]. IDO is also of key importance, as it determines the maturation of Th-type cells into regulatory suppressor Treg cells; in this way, IDO, as another element of the tumour milieu, inhibits the anti-tumour immune response. Another very important soluble factor is PGE2, which also works by stimulating typical anti-inflammatory cytokines, such as IL-10, and blocking the synthesis of certain interleukins, such as TNF-α, INF-γ and IL-12 in TAMs and DCs. PGE2 also reduces the expression of cytokines, i.e., INF-γ and IL-4, shifting the immune balance towards Th2 cells; in this way, it contributes to an increase in anti-tumour activity by promoting immunosuppressive cells such as regulatory CD4^+^CD8^+^FoxP3^+^ Tregs [98,103,116]. The mesenchymal stem/stromal cells (MSCs) can also stimulate and promote the differentiation of co-called myeloid-derived suppressor cells (MDSCs) in the bone marrow (BM). The mitogen-activated protein kinase (p38MAPK) pathway was found to be initiated in MDSCs induced by MSCs. In addition, another analysis found that the co-culture of BM cells with MSCs also resulted in phosphorylation of c-Jun N-terminal kinase (JNK) following stimulation with granulocyte-macrophage colony-stimulating factor (GMC-SF); however, the effects of MSCs on TGF-β1, TGF-β2 and IL-10 production in BM cells was abrogated by inhibition of JNK/MAPK signalling [117].

In summary, MSCs have a strong inhibitory effect on adaptive immune cells, and this ability is exploited by cancer cells. It has also been suggested that MSCs may contribute to the formation or inhibition of HNC neoplastic lesions by the promotion of tumour evolution via various mechanisms. Tumour-associated MSCs may determine the growth of HNC by various mechanisms, such as secreting biomolecules, promoting cell–cell contact, facilitating the acquisition of epithelial-to-mesenchymal transcriptional properties, suppressing the protective reactivity of immune cells, enhancing angiogenesis, or undergoing differentiation into other tumour microenvironment components such as TAMs and CAFs. The resulting changes can contribute to the acquisition of drug resistance and thus to treatment failure.

#### 2.3.2. The Innate Immune Response

The innate immune response is linked to the stimulation of Toll-like receptors (TLRs) in polymorphonuclear cells (PMNs) such as monocytes and macrophages, and in various types of epithelial cells. The TLR family, comprising TLR1 to TLR13, has been recognised in humans, and its members serve as pattern recognition receptors (PRRs). Some of these recognise molecules that are generally shared by pathogens, which are known as pathogen-associated molecular patterns (PAMPs), such as microbial-associated molecular patterns (MAMPs) expressed by pathogens and damage or death-associate molecular patterns (DAMPs) released and expressed by damaged or killed host cells. TLR activation begins through the stimulation of myeloid differentiation factor-88 (MyD88), a member of the TIR family, and the TIR domain containing an IFN-β inducing adapter (TRIF). Following this, the TLRs, except TLR3, undergo dimerisation. MyD88 signalling primarily results in the activity stimulation of nuclear factor NF-κB and mitogen-activated protein kinase (p38MAPK). MyD88 then recruits IL-1R-associated kinases (IRAK1-3), which phosphorylate and activate the molecule TRAF6; this in turn polyubiquinates the protein TAK1, which phosphorylates IKK-β. This signalisation allows NF-κB to migrate into the cell nucleus, resulting in the transcription of inflammatory cytokines and their subsequent induction [118].

In injured or tumorigenic tissues, local biomolecules such as cytokines (TNF-α, endotoxin LPS), hypoxia and Toll-like receptor (TLRs) ligands, activate MSCs to secrete growth factors such as VEGF, FGF2, IGF-1, or HGF. This stimulation takes place via an NFκB-dependent mechanism that drives tissue regeneration and angiogenesis, reduces anti-tumour immunity and allows the tumour to escape from immune surveillance in a 3-D culture system [119,120,121,122]. It is known that MSCs with mainly pro-inflammatory effects are also activated during infections and early-stage inflammatory conditions. Exposure to TLR2 (peptidoglycan) from Gram-positive bacteria or TLR4 (LPS) from Gram-negative bacteria encourages MSCs to diffuse to the site of damage and promotes immunity [123]. Thus, production of IL-6, IL-8, IFN-β, MIF and GM-CSF by MSCs may up-regulate neutrophil chemotaxis to the site of infection or injury, enhancing their activation and phagocytosis whilst promoting their survival [110,124]. For example, the activation of pro-inflammatory chemokines such as CCL2, CCL3 and CCL12 by MSCs recruits monocytes to these sites, where they differentiate into pro-inflammatory M1 macrophages (TAMs). Subsequent secretion of GM-CSF by MSCs shifts the balance towards macrophages with an M1 phenotype, thereby increasing the bacterial clearance and early wound healing responses. Also, the secretion of chemokine (C-X-C motif) ligand CXCL-9, CXCL-10, macrophage inflammatory protein MIP-1α, MIP-1β and RANTES plays an important role, as these also increase lymphocyte recruitment. It has recently been observed that this phenotype can be influenced by the media used for the in vitro expansion of MSCs: media supplemented with platelet lysate promoted a pro-inflammatory MSC phenotype and GM-CSF secretion, which may also result in the recruitment of immune cells and the maintenance of macrophages in the M1 phenotype [110,125].

Many studies have suggested that MSCs may support or inhibit HNC initiation and their evolution through various inflammatory mechanisms. By secreting biomolecules with trophic properties, tumour-associated MSCs may promote cell–cell contact, thus facilitating the epithelial-to-mesenchymal transition, suppressing the protective activities of immune cells, enhancing angiogenesis, or undergoing differentiation into other neoplastic niche components such as cancer-associated fibroblasts (CAFs). Such changes may determine the growth of HNCs, influence the acquisition of drug resistance, and thus contribute to treatment failure. Moreover, with increasing evidence indicating the function of MSCs in the direct regulation of the innate and adaptive immune system, MSC therapy may be seen as a new and promising alternative in the treatment of cancers such as HNC, as well as various other diseases. It is known that although the particular pathways by which MSCs exert their immunomodulatory properties in vivo remain largely incompletely understood, due to the idiosyncratic nature of the microenvironment and paracrine signals, they are identified to exert substantial effects on various immune cell subsets. Further understanding of the role of active factors secreted by MSCs in the secretome, as well as their interactions, will be of key importance for improving and developing new clinical protocols for MSC-based cell therapy.

A schema of MSCs’ immunosuppression mechanisms and their role in immune cell activity is shown in Figure 1B.

## 3. Studies on the Role of Mesenchymal Stromal Cells (MSCs) in Head and Neck Cancer. Effects of the Secretome on HNC. Therapeutic Potential of MSCs. The Pharmacological Strategies of MSC-Based Therapies in HNC

### 3.1. The Stemness Phenotype of Mesenchymal Stromal Cells (MSCs) in HNC. The Role of MSCs in Tumorigenesis, Progression and Drug-Resistance Mechanisms in HNC

#### 3.1.1. In Vitro Models of HNC

##### Pro-Tumour Activity of MSCs in In Vitro Models of HNC

When describing MSCs and cancer cells in the neoplastic milieu, it is necessary to mention the biomolecules present within the “secretome” [79,81]. Mesenchymal stem cells (MSCs) are multipotent cells that demonstrate significant potential in human tissue regeneration due to their capability to migrate to sites of damage, inflammation or cancer to inhibit the immune reactivity and reduce immune cell differentiation and motility. Many of these are derived from the patient’s own bone marrow (BM-MSCs) or fat tissue (A-MSCs). The crosstalk between neoplastic cells, and various lineages of MSCs, can modify the production of key biomolecules influencing neoplastic cell behaviour and other cell populations.

Interestingly, several lines of evidence indicate that MSCs lose their immunosuppressive and regenerative potency after multiple passages in vitro. Moreover, various molecules can affect the progression of HNC by modulating the behaviour of MSCs. For instance, the IL-6, platelet-derived growth factor (PDGF), and matrix metalloproteinases (MT-MMPs) such as MMP-2, MMP-9, MMP-14 and alpha-1 type I collagen (encoded by COLA1) secreted by tumour cells are known to influence MSC activity. Similarly, β-2-microglobulin (B2M), cell communication network factor 2 (CCN2), vascular endothelial growth factor (VEGF), tumour necrosis factor-beta (TNF-β), stromal cell-derived factor 1 (SDF-1), serum stem cell growth factor-beta (SCGF-β), periostin, or osteoblast-specific factor 2 (POSTN/OSF-2), CCL5, IL-6, FGF19, miR-8485 may regulate progression and control cell viability, divisions, differentiation, motility, aggressiveness, and also the epithelial-to-mesenchymal transition (EMT). Additionally, various lineages of MSCs determine the immune response, and consequently modulate HNC behaviour, by regulating the activity of IDO, CD39, and CD73 and differentiation into specific cell types, i.e., fibroblasts, chondroblasts, adipose tissue and myofibroblasts. Also, other active proteins such as Gremlin-1 (GREM1), bone morphogenetic protein 4 (BMP4), TGF-β, MT-MMPs, laminin-5, integrin, and EGFR promote cancer cell migration and invasion and EMT. Importantly, the fusion of MSCs and cancer cells provokes the secretion of the DUSP family dual specificity phosphatase 6 (DUSP6) that can regulate NO production by MAPK kinases and reduce cancer cell survival [81]. It is important to point out that transplanted MSCs do not always engraft and differentiate at the site of injury but might exert their therapeutic effects through secreted trophic signals. The most commonly secreted factors regulating tumour niche cell function are well-known ones whose activities are related to the biological effects of MSCs. These include connective tissue growth factor (CTGF), SERPINE1, TGF-β1, Dickkopf-related protein 3 (Dkk-3) and a myeloid-derived growth factor (MYD-GF). They also concern the secretion of newly identified factors whose roles are not well examined, for example, an aminoacyl-tRNA synthetase-interacting multifunctional protein-1 (AIMP1), C-type lectin domain containing 11A (CLEC11A), growth arrest specific 6 (GAS6), which regulates of natural killer cell differentiation and apoptotic cell clearance, and heparin binding growth factor (HDGF). Another compound is inhibin β-A (INHBA), which induces EMT and accelerates the motility of cancer cells by activating the TGF-β and proprotein convertase subtilisin/kexin type 5 (PCSK5) [79].

For instance, Moravcikova et al. [71] used a proteomic analysis system to distinguish issue variations in the cell surface MSC CD45^−^/CD31^−^/CD34^−^/CD73^+^/CD105^+^ antigens from native BM-MSCs through serial culture passage. The findings demonstrate that cancer cell-secreted IL-6 and PDGF in the tumour milieu are necessary to stimulate MSC migration into the head and neck tumour niche. The authors observed characteristic changes in the adipogenic and osteogenic differentiative potential during the initial expansion and invasion. The most prominent included decreases in FasL, CD98, CD205, and CD106 antigens, accompanied by a gain in the expression of CD49c, CD63, CD98, and class I/II of MHC molecules. These were accompanied by loss of MAC-inhibitory protein/CD59, loss of ICAM-1/CD54, and increase in CDKN2A expression, as well as increased CD10 expression with adipogenic and osteogenic potential. Watts et al. [126] described the secretion profile of HNSCC cells in vitro based on the JHU-011, JHU-012 and JHU-019 cell OSCC lines. The secretome included stromal cell-derived factor 1 (SDF-1 or CXCL-12), growth-regulated protein alpha (Gro-α or CXCL1), VEGF, PDGF, cytokines IL-6 and L-8, as well as PDGF-AA, as an inhibitor of the PDGF-AA receptor and PDGFR-α; these factors decreased the MSCs’ stromal chemotaxis to the oral cavity and oral pharyngeal squamous cell carcinoma (OPSCC) cells. The presence of BM-MSCs in HNSCC-derived secretory molecules increased the migration of MSCs towards cancer cells and their invasion, while these were reduced by the inhibition of IL-6 and PDGFR-α. Similarly, Kansy et al. [127] showed that when incubated in supernatants obtained from the FaDu (ATCC HTB-43) and UM-SSC-22B HNSCC cell lines, tumour-derived MSCs promote the progression of head and neck cancer stroma. This was attributed to the production of tumour-derived MSCs containing inter alia IL-1β, IL-2, IL-4, IL-6, IL-8, granulocyte-macrophage colony-stimulating factor (GM-CSF), granulocyte colony-stimulating factor (G-CSF), INF-γ, macrophage inflammatory protein 1β (MIP1β or CCL4), stromal cell-derived factor (SDF)-1α, and TNF-α in the secretome. These findings confirm that the stromal cells and tumour niche cells engage in crosstalk, resulting in enhanced HNSCC growth when xenografted into recipient animals in vivo. Ji et al. [128] reported a similar secretome profile among MSCs from gingival-derived normal tissue (GMSCs) within the tumour microenvironment from oral cell lines (CAL-27 and WSU-HN6) after in vitro supplementation with anti-inflammatory IL-10. The GMSCs were found to influence the oral cancer cells via direct co-culture and indirect co-culture systems. A direct co-culture cell proliferation assay indicated that GMSCs inhibited the growth and invasion phenotype of oral cancer cells. The conditioned medium derived from the GMSCs (GMSCs-CM) also exerted an anti-cancer effect, which indicates that soluble factors in GMSCs-CM play a key role in GMSC-induced inhibition of cancer cell growth. Additionally, the study confirmed that GMSCs could act as activators of tumorigenesis through the up-regulation of pro-apoptotic and cell death proteins, including p-JNK, cleaved PARP, cleaved caspase-3 and Bax, and the down-regulation of proliferation- and anti-apoptosis-related proteins such as p-ERK1/2, Bcl-2, CDK4, cyclin D1, PCNA and survivin. Scherzed et al. [129] also obtained interesting data on secretome-based pro-cancer mechanisms in an HNSCC HLaC78 cell line. The study investigated whether human mesenchymal stroma cells (hMSCs) support cell motility and cytokine secretion. Interestingly, hMSCs enhanced FaDu and HLaC78 cell invasiveness. Cancer cell motility was increased by cytokines such as IL-6, IL-8 and VEGF. Moreover, the inhibition of IL-6 in the MSC secretome decreased HNSCC cell proliferation, which was partly dependent on the MAPK/ERK signalling pathway. Similarly, exposure of human tongue squamous cell carcinoma (TSCCa and CAL-27) cell lines to the MSC secretome resulted in a significant increase in CCN2 in BM-MSCs. However, in tumour-derived MSCs, only CCN2 inhibited cancer cell proliferation, mobility and invasion, and it decreased the levels of MMP-9, MMP-2 and epithelial-mesenchymal transition markers in vitro. It is also not surprising that higher expression of CCN2 and the connective tissue growth factor (CTGF) was noted in HNSCC tissues than in normal adjacent non-cancerous tissues, and this may contribute to the higher aggressiveness of TSCC cells via the promotion of tumour development [130]. Liu et al. [131] also analysed the mechanisms of BM-MSCs involved in promoting the development, progression, invasion, and metastasis of head and neck cancer cells (CAL-27 and HM-4) and the tumour-promoting role of periostin (POSTN) in HNC. In vitro data derived from HNC cells cultured in the presence of BM-MSC-conditioned medium (MSC-CM) indicated that the stem cell determines cancer progression by increasing cell proliferation, migration, and epithelial–mesenchymal transformation (EMT), and by blocking apoptosis and altering the expression of proteins regulating the cell cycle. Most importantly, BM-MSCs promoted HNC aggressiveness through the PI3K/AKT/mTOR signalling pathway, which was mediated by POSTN. As previously mentioned, IL-6 present in the MSC secretome, or produced directly in the tumour milieu, may promote EMT and the acquisition of epithelial stem-like cell properties [132,133,134,135]. An in vitro study of an ameloblastoma epithelial cells (AMs), an aggressive odontogenic neoplasm carcinogenesis model, by Jiang et al. [132] confirmed increased levels of the pro-tumorigenic and pro-angiogenic cytokine IL-6 in the supernatants from isolated mesenchymal stromal cell (AM-MSC) culture. The supernatants inhibited cell proliferation, promoted differentiation, and inhibited epithelial differentiation of the epithelial cells (AM-EpiCs) from follicular AMs, thus increasing pre-malignant lesions and accelerating the process of carcinogenesis by chemical carcinogenesis. The secretome of the ameloblastoma-derived MSCs, which contained angiogenic IL-8, functioned through the MAPK/STAT3/SLUG signalling pathways and SNAIL-l, Vimentin and ZEB1 factors. Another work on oral mucosal MSC-derived exosomes (OM-MSC-EVs) and their potential therapeutic target in oral premalignant lesions and OSCC cancer cells was presented by Li et al. [133]. The findings indicate that the proliferation and migration of oral leucoplakia with dysplasia mesenchymal stromal cells (LK-MSC) were down-regulated compared with normal oral mucosa (N-MSC) and oral carcinoma (Ca-MSC) cells. It can also be seen that the exosomes secreted by LK-MSCs play an important and essential role in promoting proliferation, migration, and invasion in vitro. Interestingly, microarray analyses of MSC-derived exosomes confirm the presence of microRNA-8485 (miR-8485) in the MSC-derived exosomes. Exosomal miR-8485, present in both leucoplakia and cancer cells, enhanced the proliferation, migration and invasion of cancer cells under in vitro co-culture conditions. Shi et al. [136] analysed the probable factors determining the progression of oropharyngeal premalignant lesion to NPC carcinoma in vitro. It was found that the isolated BM-MSC-EVs significantly regulate fibroblast growth factor-19 (FGF-19); they therefore act as a potent regulators of nasopharyngeal carcinoma cell lines (CNE1, CNE2, 5-8F and 6-10B) via the FGF19-FGFR4-dependent ERK signalling cascade and by modulating EMT.

Tumours of the oesophagus, such as squamous cell carcinomas (OESSCs), were also included in the article due to their similar histology to the group of the most common head and neck squamous cell carcinomas (HNSCCs) and for comparison of the research results concerning MHCs. For instance, Wang et al. [137] also identified B2M in the secretome of BM-MSCs from oesophageal TE1 and Eca109 cell lines. This interesting study used B2M-encoding gene knockdown to demonstrate that the gene played a part in the invasion and migration of tumour cells. Another interesting in vitro study by He et al. [138] found that Gremlin-1 overexpression markedly promoted the proliferation and invasion of human oesophageal squamous cell carcinoma in ECa109 and TE-1 cell lines and xenograft tumour models. In addition, shRNA silencing of GREM1 mRNA in MSCs (shGREM1-MSCs) inhibited and then reversed the increased malignancy of OESCC, and medium conditioned with shGREM1-MSCs (shGREM1-MSCs-CM) blocked the cell cycle process and cell invasion in vitro. The experimental shGREM1-MSCs-CM-induced anti-tumour stem cells effects seem to be controlled by the TGF-β/BMP4 (transforming growth factor-β/bone morphogenetic protein-4) signalling pathway, which was also associated with a decrease in TGF-β and Smad-2 and Smad-3 activity, and with an increase in BMP4, Smad-1, Smad-5 and Smad-8 expression. Nakayama et al. [139] reported an increase in aggressive phenotype and pro-tumorigenic interactions between adipose-derived MSCs (A-MSCs) and EC-GI-10 (well-differentiated type) and TE-9 (poorly differentiated type) OESCC cell lines in vitro. Pro-neoplastic activity was positively associated with the expression of phosphorylated-insulin-like growth factor-1 receptor (p-IGF1R) and negatively associated with the human epidermal growth factor receptor 2 (EGFR-2) in OESCC cancer tissues. The authors suggest that co-culture of A-MSCs and cancer cells may be a pro-tumorigenic factor promoting neoplastic invasion and increasing the level of MMP-9 and laminin. Similar data were also presented by Wang et al. [140], who reported that fusion of human umbilical cord-derived mesenchymal stem cells (UC-MSCs) with human oesophageal carcinoma cell lines (EC9706) noticeably blocked the carcinogenesis of OESCC. A comparison of the gene expression profiles of human mesenchymal stem cells, oesophageal cancer cells and hybrids indicated that the OECs-hMSC fusion induced apoptosis and benign transdifferentiation. Moreover, fusion also strongly increased the activity of dual specificity phosphatase 6 (DUSP6)/mitogen-activated protein kinase phosphatase-3 (MKP-3), the key regulators in the p38MAPK pathway, and exogenous overexpression inhibited tumour growth.

##### Anti-Tumour Activity of MSCs in In Vitro Models of HNC

Some in vitro analyses have found mesenchymal stromal cells to have the opposite effects on a number of pathways associated with cancer processes in the head and neck region [128,141]. For instance, Ji et al. [128] found MSCs derived from normal gingival tissue (G-MSCs) to inhibit the proliferation of oral squamous cancer cells (CAL-27 and HN6 OSCC cell lines) in vitro and in vivo. MSC-EVs from the secretome present in the studied co-culture systems down-regulated OSCC cells by inducing neoplastic cell death and blocking proliferation. Interestingly, the G-MSC secretome down-regulated the expression of genes associated with proliferation and anti-apoptosis activity, such as p-ERK1/2, Bcl-2, CDK4, Cyclin D1, STAT3, PCNA and survivin, and the ERK signalling pathways; it also up-regulated the JNK cascade and expression of pro-apoptotic genes, including JNK, cleaved PARP, and cleaved caspase-3, which negatively regulated the cell cycle and tumour proliferation and increased angiogenesis. Moreover, treatment with MSC secretome blockade and JNK signalling inhibitor increased cancer cell proliferation in vitro. The dual role of human MSCs in tumour cell growth, mainly their anti-cancer effect, was also presented by Li et al. [141]. This in vitro study analysing the effect of the BM-MSC secretome in oesophageal squamous cell carcinoma cell lines (Eca-109) found that hMSC-conditioned medium repressed the proliferation and invasion of Eca-109 cells, arrested the cell cycle in the G1 phase and intensified the apoptosis of OESCC in vitro in a co-culture system. Treatment with the conditioned medium also reduced the expression of PCNA antigen, cyclin E, pRb protein, Bcl-2, Bcl-xL and MMP-2, and it blocked the formation of cyclin E-cyclin 2 (CDK2)-dependent kinase complexes.

##### Modulation of Immune and Inflammatory Cells by MSCs in In Vitro Models of HNC

Several important studies have also found MSC-mediated immunomodulation to have pro- or anti-tumorigenic potential in HNC models [142,143,144,145]. For example, Liotta et al. [142] found that HNSCC-derived MSCs inhibited the proliferation of CD4^+^ and CD8^+^ T cells and promoted the down-regulation of INF-γ and TNF-α expression. Interestingly, mesenchymal cells isolated from tumours co-expressed CD29, CD105, and CD73, but not CD31, CD45 and CD133; they also presented human epithelial antigen like bone marrow-derived MSCs (BM-MSCs). Furthermore, HNSCC-isolated MSCs were also characterised by significant immunosuppressive activity on in vitro stimulated T cells, mainly mediated by indoelamine 2,3-dioxygenase (IDO) activity. Moreover, the abundance of cancer-derived MSCs was directly correlated with the tumour volume and inversely with the frequency of tumour-infiltrating leukocytes (TILs). Similar conclusions were also presented by Mazzoni et al. [143], who highlighted the involvement of MSC IDO-1 in the immunosuppression of the proliferation of HNSCC-derived MSC-mediated T cells in an HNC model. Also, MSCs derived from head and neck cancers inhibited the function and proliferation of T lymphocytes and suppressed the T cell immune response via the down-regulation of amino acid oxidase, known as IL-4 induced gene 1 (IL4I1), the catabolic products such as H_2_O_2_, and the kynurenines activation detected in various types of cancer cells. The study also demonstrated that neutralisation of IL4I1 activity can block tumour cell migration and restore effective anti-tumour immunity. Another interesting study by Schuler et al. [144] investigated the effect of CD39 and CD73 expression in HNSCC-derived MSCs generated from tumour tissue and autologous MSCs from healthy control tissue. It proposed that the conversion of extracellular ATP (eATP) to immunosuppressive adenosine (ADO) by the functionally active ectonucleotidases CD39 and CD73 constituted an immunosuppressive mechanism used by hematopoietic immune cells. Furthermore, MSCs from tumours demonstrated lower CD39 and CD73 protein expression compared to non-cancerous tissue, and this expression correlated with decreased ATP metabolism and the suppression of CD4^+^ T-cell proliferation. Therefore, CD39 and CD73 may also constitute a potential novel checkpoint inhibitor of targets due to their tumorigenic action [146]. Allard et al. reported that in response to conditions typically occurring in neoplastic disease, such as hypoxia, various cells in the tumour microenvironment acquire adenosine-generating capabilities; these include cancer cells, endothelial cells, CAFs, CD4^+^CD25^+^Foxp3^+^ Tregs, Tr1 cells, Th17 cells, γδ T cells, NK cells, invariant cells (i)NKT, effector and memory T cells, B regulatory cells (Breg), myeloid-derived suppressor cells (MDSC), macrophages and neutrophils. In turn, the described molecular mechanism increased the survival of tumour cells and metastases, promoted angiogenesis, increased fibrosis, and up-regulated the suppressive function of Tregs, Tr1, TAMs and MDSCs; by doing so, it also promoted antigen tolerance, inhibited the effector function of lymphocytes and prevented the differentiation of memory T cells into effector cells, facilitating tumour growth. Therefore, the authors predicted that adenosinergic and other purinergic-targeting therapies may have clinical application, and their development in combination with other anti-cancer modalities may result in promising future therapeutic approaches. Similar conclusions were presented by Rowan et al. [147], who demonstrated that the use of human adipose tissue-derived stromal cells (A-MSCs) promotes the migration and early metastasis of human CAL-27 and SCC-4 head and neck cancer cell lines and NUDE mouse xenografts. The authors observed that MSCs create an inflammation-induced and tumour-friendly microenvironment through down-regulated expression of CD73 and metabolism of ATP, which inhibited T cell proliferation and activity among CD4^+^ and CD8^+^ lymphocytes, and induced TAM M1 polarisation and higher Treg cell immunosuppressive function.

##### Therapeutic Potential of MSCs in In Vitro Models of HNC

There is also great interest in the chemopreventive and therapeutic potential of secretome components and MSC activity in HNSCC cancers, one of the most common malignancies of the head and neck area [127,128,129,130,131,132,133]. Importantly, several recent studies have reported that biomolecules released from the secretome to the neoplastic niche may not only determine the proliferation, growth and invasiveness of head and neck tumours but also determine death resistance. Most importantly, a therapeutic strategy has been proposed in which MSCs obtained from different tissues can be loaded in vitro with anti-cancer drugs [148].

For example, MSCs have been isolated and expanded from gingival papilla (GinPa-MSCs) and infused with three important anti-neoplastic drugs: Paclitaxel (PTX), Doxorubicin (DXR) and Gemcitabine (GCB) [148]. The results clearly demonstrate that GinPa-MSCs efficiently absorbed these chemotherapeutics and then expelled them into the tumour milieu in their active form. The drugs were delivered in specific amounts intended to produce the stem cell growth factor-beta (SCGF-β), which inhibits proliferation of human SCC154 oral cell line growth in vitro. Also, Wang et al. [149] found bone marrow mesenchymal stem/stromal (BM-MSCs) cells to have anti-apoptotic effects when co-cultured with human-derived oropharyngeal squamous carcinoma JHU-12 and JHU-019 (OPSCC) cells. This phenomenon was associated with the activation of PDGFR-α/AKT mediated signalling pathways. This paracrine-mediated signalling highlighted the chemotaxis of MSCs in OPSCC. Moreover, the enhancement of the PDGFR-α/AKT pathway by MB-MSCs promoted the expression of anti-apoptotic Bcl-2 and decreased the sensitivity to cisplatin. However, OPSCC-derived JHU-012 cells grown in co-culture with MSCs were significantly more susceptible to CDDP following pretreatment with the receptor tyrosine kinase Crenolanib, a PDGFR-α inhibitor, compared to cancer cells grown alone. Another interesting study was conducted by Liu et al. [150], who showed that isolated BM-MSCs actively interacted with HNSCC cancer cells in vitro (SCC-25 cells) and in vivo, and this interaction intensified the key mechanisms responsible for tumour progression and growth and drug chemoresistance. Parental head and neck cancer cells, either fused with MSCs or exposed to MSCs, were orthotopically transplanted into the tongues of mice. The fused cancer cells demonstrated more intense mesenchymal cell features, i.e., higher expression of POSTN, GDF11, IGFBP5 and CXCL11, and down-regulation of DAPK1, as well as greater proliferation and viability. Moreover, the HNSCC cells incubated with MSC were associated with a more aggressive course of neoplastic disease compared to the parental cell line. Interestingly, a key condition for the transmission of signals from growth factor receptors to regulate gene expression and prevent apoptosis was found to be the PI3K/PTEN/AKT signalling cascade. Importantly, all the HNSCC cell lines exposed to MSCs developed resistance to Paclitaxel, which persisted for up to 30 days after the initial co-incubation period. Interesting conclusions were also presented by other researchers, who confirmed the involvement of MSC-induced collagen and GREM-1 in neoplastic tissue and the immunosuppression of HNSCC-derived MSC-mediated T memory cells in an HNC model [151,152,153].

Taken together, these findings suggest that selective inhibition of MSC function in the TME, or the blockade of key signalling pathways for their activity, may constitute a viable treatment strategy for combating tumorigenesis and chemoradioresistance; however, such a development requires further mechanistic and translational research in head and neck cancers.

##### Limitations of In Vitro Models of HNC

However, although in vitro models are valuable for obtaining new information, they cannot fully mimic the complex tumour microenvironment. In contrast, in vivo or animal models such as mouse xenograft models have an advantage in that they can mimic the tumour niche and key intercellular interactions, i.e., communication with stromal cells, stem cancer cells and cells of the innate and adaptive immune system. Such limitations of in vitro studies should always be taken into account when drawing final conclusions regarding the role of mesenchymal cells in various human cancers, including head and neck cancers. Nevertheless, in vitro studies offer the advantage that the analysed cells are exposed to a relatively homogeneous environment. This affords the researcher ample opportunity to study the effects of constant oxygen levels, induced hypoxia, nutrient composition and a conditioned medium of MSCs (MSCs-CM), as well as limited interactions with other cells.

Table 1 summarises selected in vitro studies regarding the role of MSCs in the tumour microenvironment included in this review and their findings.

#### 3.1.2. In Vivo and Animal Models of HNC

##### Pro-Tumour Activity of MSCs in In Vivo and Animal Models of HNC

Various animal and in vivo models have also been used to explore the role of MSCs in the development and progression of head and neck squamous cell cancer (HNSCC). Mouse and hamster studies have found mesenchymal stromal cells to regulate the initiation and growth of HNC and its lymph node and distant metastases [152,154,155,156,157,158]. For instance, Liu et al. [131] investigated whether bone marrow mesenchymal stem cells (BM-MSC) are recruited to the tumour microenvironment and have tumour-promoting effects in a murine model of HNC carcinogenesis induced with periostin. BM-MSC was found to promote tumour development, invasion, degree of aggressiveness, lymph node metastases and shorter survival. It was related to enhanced expression of POSTN and the epithelial–mesenchymal transition (EMT) in cancer tissues. The POSTN mRNA level was also higher in CAL-27 cell lines of BM-MSC-HNSCC tumours, which was associated with a high pathological grade, proliferation rate, tumour volume and lymph node metastasis. The researchers suggested that their findings were dependent on the activity of important molecular pathways, such as POSTN-mediated PI3K/AKT/mTOR signalling and N-cadherin activity. Another interesting in vivo analysis by Meng et al. [158] evaluated the potential of interactions between tumorous cells obtained from surgical resection, normal oral cells and their surrounding stromal microenvironment to induce tumorigenesis and progression for oral squamous cell carcinoma (OSCC). The study also analysed the potential targets for therapeutic intervention for OSCC. The data indicated that tumour formation in CG2, HSC-2, and Tca8113 cells infected with lentivirus expressed enhanced the levels of TGF-β receptor III (TβRIII), and this molecule was an important potential epithelial–mesenchymal common target. A recent research study by Jiang et al. [132] determined that MSC-derived IL-6 contributes to the pathogenesis and progression of ameloblastoma (AM). Interestingly, both in vivo and in vitro studies on fresh tumour samples confirmed that AM-MSC-derived IL-6 enhanced the levels of EMT factors and stem cell-related genes in epithelial cells from follicular AM (AM-EpiCs). Furthermore, the biological actions of the mesenchymal stromal cells were stimulated via the STAT3 and ERK1/2-mediated signalling pathways or by the SLUG gene. The researchers additionally noted that the growth of AM was inhibited by a specific inhibitor of STAT3 or ERK1/2, or by knockdown of SLUG gene expression; this appeared to have the effect of down-regulating the expression of EMT- and stem cell-related genes in AM-EpiCs.

This article also includes the in vivo and animal model analyses of carcinomas of the oesophagus and the nasopharyngeal region due to their similar squamous cell histology to the head and neck squamous cell carcinomas (HNSCCs). Interesting observations regarding the pro-tumorigenic potential of MSCs were presented by He et al. [138], who examined the effect of inhibition of SALL4 that reduces tumorigenicity involving epithelial-mesenchymal transition via Wnt/β-catenin pathway in esophageal squamous cell carcinoma. The results revealed that SALL4 might serve as a functional marker for esophageal carcinoma cancer stem cell, a crucial marker for prognosis and an attractive candidate for target therapy of OESCC. Also, Shi et al. [136] analysed the effect of bone marrow mesenchymal stem cell-derived exosomes (BM-MSC-EVs) in the development and progression of nasopharyngeal carcinoma (NPC) in a model of female NOD/SCID mice subcutaneously inoculated with NPC CNE1 and CNE2 cells to induce cancer. It was found that activation of the EMT markers and stimulation of the fibroblast growth factor (FGF19-FGFR4)-dependent ERK signalling cascade resulted in the greatest facilitation of proliferation, migration and tumorigenesis.

Other publications have examined the effects of various carcinogens in MSC-dependent animal HNC carcinogenesis models. These have confirmed the modulatory pro-tumorigenic effect of MSCs, which play a significant role in tumour progression, metastasis, and cancer recurrence, further supporting their potential role in targeted cancer prevention [154,159]. For example, Chen et al. [154] proposed that various mesenchymal stem cells of different origins, such as normal mucosa-derived MSCs (N-OMSC), dysplasia-derived MSCS (D-OMSC), cancer-derived MSCs (C-OMSC), and the corresponding BM-MSCs, may be involved in tumour formation in oral carcinogenesis by inhibiting T CD3^+^ and CD45^+^ cell numbers and proliferation. The experimental carcinogen 4-nitroquinoline-1-oxide (4NQO) initiated dysplasia and cancerous lesions in the oral cavity of a female Sprague Dawley rat OSCC model. The suggested cause of the pro-tumour activity was an increase in the proportion and proliferation capacity of oral lesion-derived MSCs, which effectively reduced the proportion of T immune cells and significantly immunosuppressed their activity associated with oral mucosa malignancy. Furthermore, increased expression of chemokines CCL21 and CXCL12, and SDF1 was noted in the secretome from cancer tissue-derived MSCs. Interesting results were also reported by Kumar et al. [159], who analysed the expression of adipokine, chemerin (RARRES-2) and its receptor (ChemR23) in myofibroblasts (CAMs) and other squamous cell oesophageal cancer stromal cells, and they examined their role in recruitment of bone marrow-derived MSCs and tumour progression. The results of the in vitro experiment and xenograft model indicated that chemerin stimulation of MSCs enhanced the phosphorylation of p42/44 and p38, as well as JNK-II kinases and their inhibitors, and PKC reversed chemerin-stimulated MSC migration. Moreover, in a xenograft model consisting of OE21 oesophageal cancer cells and oesophageal squamous cancer-associated myofibroblasts, CCX832 was found to inhibit the homing of intravenously administered MSCs. The researchers concluded that RARRES-2 secreted by CAMs constituted a potential chemoattractant for MSCs and its inhibition may delay tumour progression.

##### Anti-Tumour Activity of MSCs In Vivo and Animal Models of HNC

However, several publications fail to confirm that MSCs have an anti-tumour effect in HNCs [128,156,157,160,161]. For instance, Ji et al. [128] demonstrated that conditioned medium derived from GMSCs (GMSCs-CM) showed a strong anti-cancer effect through inhibiting the growth of OSCC cells. The authors analysed the role of MSCs derived from normal gingival tissue (MSCs-GMSCs) in regulating the proliferation and growth of oral cancer (OSCC) cells in an animal model of male BALB/C nude mice and by direct co-culture and indirect co-culture systems in vitro. Furthermore, it was also confirmed that the intracellular mechanisms responsible for inhibiting tumour growth were related to increased levels of pro-apoptotic genes, including JNK, cleaved PARP, cleaved caspase-3 and Bax, and decreased proliferation and reduced expression of anti-apoptosis-related genes such as ERK1/2, Bcl-2, CDK4, cyclin D1, PCNA and survivin. Similar conclusions were presented by Bruna et al. [156], who applied multipotent stromal cells at the precancerous stage of oral squamous cell carcinoma (OSCC) in Syrian hamsters after topical application of the mutagen 7.12-dimethylbenz-alpha-anthracene (DMBA) in the buccal pouch. The authors noted that the allogeneic bone marrow-hamster-derived MSCs (BM-MSCs) prevented oral carcinogenesis via inhibition of cancer growth and epithelial dedifferentiation. Moreover, the local administration of mesenchymal cells into the hamster oral mucosa reduced the tumour mass and volume, showed anti-proliferative (Ki-67) and pro-apoptotic (caspase 3 cleaved) activation, inhibited angiogenesis (ASMA) and decreased local inflammation (leukocyte infiltration) and differentiation (CK1 and CK4) in animals treated with MSCs compared to untreated ones; it also down-regulated the activation of pro-tumoral gene expression in precancerous lesions. Interestingly, the same team also studied the anti-cancer effect of systemic intracardial administration of allogeneic BM-MSCs with regard to the initiation and further development of precancerous conditions of OSCC; in this case cancer was induced in Syrian golden hamsters by topical application of DMBA in a single buccal pouch [157]. The authors observed that precancerous lesions progressed from hyperplasia to dysplasia, from dysplasia to papilloma, and from papilloma to carcinoma within four weeks; however, in animals injected with low and intermediate MSC doses, this process was not initiated or up-regulated by systemic administration of MSCs at the hyperplasia or dysplasia stages. All the animals treated with MSCs developed OSCC after 13 weeks of treatment, and this condition remained dependent on high doses of mesenchymal cells. Moreover, hamsters receiving BM-MSCs at the hyperplasia plus dysplasia lesion stage and the papilloma stage were significantly less likely to develop OSCC than the control animals. The researchers concluded that the injection of low and medium systemic doses of allogeneic MSCs, administered in the early stages of oral carcinogenesis, did not aggravate the progression and growth of precancer lesions. However, further tumour growth was associated with high doses of BM-MSCs in the later stages of OSCC, and this was related to the presence of persistent chronic inflammation and the intensification of immunosuppressive phenomena inhibiting anti-tumour defence mechanisms. Also, Tan et al. [160] analysed the tumorigenic potential in mesenchymal stem/stromal cell-derived small extracellular vesicles (MSC-sEV) in athymic nude mice with FaDu human head and neck cancer xenografts using immortalised E1-MYC 16.3 human ESC-derived mesenchymal stem cells. Interestingly, the intraperitoneal injection of immortalised MSCs transformed with a proto-oncogene MYC did not appear to have a pro-tumorigenic role in initiation or anchorage-independent growth at pre- or post-exosome production of HNC tumours in an animal model. Moreover, the selected exosome production method did not affect cell growth and did not contribute to the generation of tumour-promoting MSC exosomes. Nevertheless, immortalising MSCs for exosome production may allow the production of safe exosome preparations for therapeutic applications, but further extensive research is needed.

At the end of this section, it is worth highlighting that in vitro research or in vivo studies on knockout mice and transgenic mice and hamsters provides key information on the importance of mesenchymal stem cells (MSCs) in classical target tissues. They also highlight the role of MSCs in HNC initiation, growth and development, including their effects on cancer progression, carcinogenesis and immunomodulation. The vast majority of recent data suggest that the interaction between tumour cells and MSCs within the tumour niche plays a significant role in tumour expansion and nodal or distant metastases, and thus might be exploited for therapeutic intervention. However, further studies in larger cohorts with standardised experimental protocols are needed to confirm this. It should be noted that mouse xenograft models can accurately imitate the tumour microenvironment occurring in real conditions. Such a “natural” cancer niche allows us to observe real, important interactions and communication with stromal cells, cancer stem cells and immunocompetent cells, allowing us reliable and practical conclusions to be obtained.

##### Limitations of In Vivo and Animal Models of HNC

Unfortunately, important limitations regarding the observations and results of this type of research must be taken into account. These may result from the use of MSCs from different sources, including HNSCC of different origins, as well as different or heterogeneous experimental protocols. Furthermore, in in vivo studies, bias and alternative conclusions may also arise from inter alia the heterogeneity of patient/animal samples, insufficient sample sizes of patient and control comparison groups, short post-treatment periods or variable follow-up times. Some may also fail to take into account smoking addiction and excessive alcohol consumption in patients with HNSCC. Also, the studies may be based on different populations from heterogeneous ethnic groups with varying degrees of risk of carcinogenesis in the head and neck region and who may be exposed to different environmental carcinogens. Additionally, many studies use different analytical endpoints, demonstrate fundamental differences in methodological standardisation and employ different research methods. Such variation can result in inconsistent data, even when the same mesenchymal cells are used, and this can limit the possibility of generalising the final results.

Table 2 presents selected animal/in vivo studies on MSCs in the tumour microenvironment described in this review and the data collected from them.

### 3.2. The Pharmacological Strategies of MSC-Based Treatment for Human Tumours. MSCs as Carriers of Anti-Tumour Therapeutic Biological Compounds and Their Clinical Application for Oncological Therapy

Over the past decade, research has focused on the potential use of pleiotropic mesenchymal stem cells (MSCs) as highly specialised “Trojan horses” that can deliver biological anti-tumorigenic molecules, interleukins and agents of interferons, drugs or prodrugs to primary tumour cells or the tumour milieu or metastatic tumours. The research has attracted great interest, primarily due to the fact that MSCs have an innate and induced ability to migrate to the cancer environment or convert MSCs to other cancer-associated cells in the tumour niche. The most common and promising strategies for the production of MSCs are based on genetic engineering: such approaches could transfer various types of therapeutics or biological agents to the TME or directly to neoplastic lesions, thus inhibiting early initiation or further development of the tumour. Thanks to the constant improvement in genetically manipulated MSCs, the last decade has seen very rapid development of cell therapies using various derived mesenchymal cells for oncological applications. An increasing number of preclinical and realised clinical phase I and II studies in various human cancers indicate that these specific pluripotent cells may have potential importance in personalised cell therapies because they can be easily obtained through minimally invasive procedures and then rapidly scaled up [52,162,163].

To date, the ClinicalTrials.gov website (data from ClinicalTrials.gov., U.S. National Library of Medicine, accessed on 30 April 2024) lists over fifty registered preclinical and clinical trials that have used MSCs in the treatment of various cancer diseases. Among these studies, only one, a phase I study (NCT0207932), analysed the involvement of MSCs as a therapeutic agent for the direct treatment of head and neck cancer. However, it should be emphasised that the remainder of the works, of varying methodologies, focus on the use of MSCs of various origins for treating irradiation-induced salivary dysfunction, such as hyposalivation and xerostomia/dry mouth in patients with head and neck cancers (for instance, PROSPERO CRD42021227336, NCT04489732, NCT047765392, NCT03874572, MESRIX-SAFE, MESRIX, MESRIX-II and MESRIX-III studies). The above-mentioned works will be discussed in the next section [164,165,166,167,168,169,170,171,172,173,174,175,176].

#### 3.2.1. Adenoviral Vectors and Oncolytic Adenoviruses

Several preclinical studies have been performed on other human cancers and neoplastic lesions. For example, human umbilical cord-derived MSCs (UC-MSCs) transduced with adenoviral vectors expressing IL-18, IFN-β and other key cytokines, such as TRAIL (TNFSF10), as well as key anti-angiogenic agents, pro-apoptotic proteins and growth factor antagonists, effectively inhibited tumour cell proliferation, cancer initiation and development and the formation of metastases; they were also found to induce apoptosis [52,177,178]. Similarly, genetically engineered TRAIL-expressing adipose-derived mesenchymal stem cells (A-MSCs-TRAIL^+^) created by lentiviral transductions have also shown potent anti-tumour effects in various cancer types, such as glioblastoma, hepatocellular carcinoma and haematological malignancies, such as acute lymphocytic leukaemia, or chronic myelogenous leukaemia [179,180,181,182]. In addition to cytokines and suicide proteins, several other proteins that inhibit carcinogenesis have also been used in anti-cancer engineering of MSCs. For example, MSCs with positive expression of bone morphogenetic protein 4 (BMP-4) and phosphatidylinositol 3,4,5-triphosphate-3-phosphatase (PTEN) also effectively inhibited tumour growth, induced cell cytotoxicity, and significantly prolonged survival in mouse models of various human cancers [181,183,184].

Interestingly, oncolytic adenoviruses (Ads) also have potential applications in cancer therapy due to their ability to replicate and induce programmed cell death in cancers. Unfortunately, their clinical use has been severely limited due to the lack of use of effective cell-based Ads delivery systems that could protect the transferred molecules from attack by immunocompetent cells, thus preventing virus clearance through antibody neutralisation [181]. One particularly interesting study reported the use of polymeric nanoparticle-engineered human adipose-derived mesenchymal stem cells (hA-MSCs) overexpressing the cancer-specific TNF-related apoptosis-inducing ligand (TRAIL) to target tumours in mice. The authors observed that following transplantation of patient-derived orthotopic tumour xenografts to a mouse model, engineered MSCs expressing suicide protein TRAI exhibited long-range directional migration towards tumours in patient-derived GBM orthotropic xenografts; they also showed significant inhibition of neoplastic growth and induction of apoptosis, thus reducing the occurrence of microsatellites, and prolonging animal survival [182]. An interesting concept in anti-cancer therapy is the “loading” of MSCs with other types of oncolytic viruses, which are chemotactic to various tumour cells. Such oncolytic virotherapy represents another promising alternative and effective anti-tumour therapeutic role for mesenchymal cells. One study examined the use of MSCs infected with an oncolytic adenovirus, e.g., ICOVIR5 (Celyvir) in the treatment of a murine CMT64 cell line, syngeneic for human lung cancer, as a human adenovirus-semi-permissive tumour model. The researchers found mouse Celyvir (mCelyvir) to demonstrate a significant homing capacity to CMT64 tumours. Interestingly, the combined treatment based on mCelyvir and intratumoural injections of ICOVIR5 was found to act by inhibition of neoplastic growth and induction of CD4^+^ and CD8^+^ T cell recruitment to the tumour microenvironment [185]. Another interesting study examined the use of menstrual blood-derived mesenchymal stem cells (MenSCs) infected with the CRAd5/F11 chimeric oncolytic Ads and transplanted in a mouse tumour model. This novel virus delivery platform inhibited cancer progression in a subcutaneous mouse xenograft model of human colorectal cancer [186]. Also, stem cell-released variants of the oncolytic herpes simplex virus (MSC-oHSV) demonstrated noticeable therapeutic efficacy in human advanced brain melanomas with metastatic processes in a relevant immunocompromised and immunocompetent mice tumour model. Interestingly, the authors found that intracarotid administration of MSC-oHSV effectively targeted metastatic neoplastic lesions and significantly prolonged the survival of tumour-bearing C57BL/6 mice. Moreover, the combination of MSC-oHSV with anti-PD-L1 immunotherapy also increased the abundance of tumour-infiltrating IFN-γ-producing CD8^+^ T cell subpopulation and was associated with a significant increase in the median survival of treated animals [187].

#### 3.2.2. MSCs Carrying Anti-Cancer Payloads and Drugs

Another modern method for treating animal or human cancers is the use of MSCs carrying anti-cancer payloads and drugs [52,188,189,190,191]. This technique takes advantage of the fact that MSCs have been found to effectively incorporate drugs and then release them in an active form and in sufficient amounts to inhibit various squamous cell carcinomas in vitro, including HNC. For example, conditioned medium with MSCs originating from gingival papillae (GinPa-MSCs), which had been treated with Paclitaxel (PTX), Doxorubicin (DXR) and Gemcitabine (GCB), was found to be effective against a line of tongue squamous cell carcinoma (SCC154). The authors emphasised that compared to other sources of MSCs, acquiring GinPa-MSC is minimally invasive, and the stem cells can be easily expanded and effectively loaded with anti-cancer drugs, establishing an effective “cellular drug delivery system”. Moreover, drug-loaded gingival mesenchymal stromal cells, particularly with a cargo of GCB, significantly hindered the growth of, and showed anti-proliferative effects against, a tongue squamous cell carcinoma SCC154 cell line. This was also accompanied by significantly higher expression of hENT1, the main carrier involved in the transport of gemcitabine in cancer cells. The study indicated that such anti-cancer strategies may be used as a basis for future application in oral oncology [188].

#### 3.2.3. Nanoparticles

Recent studies have also examined the development of therapeutic strategies aimed at improving the loading capacity and efficiency of MSCs. One promising approach to increasing the anti-cancer effectiveness of MSCs loaded with anti-cancer drugs is based on nanoparticles. Recent studies clearly indicate that drug-encapsulated nanoparticles offer many therapeutic benefits, such as the ability to accumulate in tumour tissues or the neoplastic milieu, mitigate the non-specific toxicity of anti-cancer drugs, prevent sudden uncontrolled release and limit side effects [189,190,191]. In one study, engineered mesenchymal stem cells with drug-loaded nanoparticles carrying Paclitaxel (PTX) were tested against an A549 orthotopic lung tumour model. Despite the use of much lower doses of PTX, the nanoengineered MSCs significantly inhibited tumour growth and improved the survival of immunocompetent C57BL/6 albino female mice bearing orthotopic Lewis lung carcinoma (LL/2-luc) [189]. Similar observations were obtained from MSCs loaded with poly(D,L-lactide-co-glycolide) (PLGA) nanoparticles encapsulated with Paclitaxel (PTX-PLGA) for orthotopic glioma therapy in male Sprague Dawley rats. It was found that the MSCs initiated with PTX-PLGA nanoparticles demonstrated significantly greater and prolonged release of PTX in the form of free nanoparticles compared to those initiated with PTX. Moreover, transfer of Paclitaxel from MSCs to the tumour mass strongly induced neoplastic cell apoptosis in vitro. Furthermore, the animals demonstrated significantly longer survival after implantation of PTX-PLGA nanoparticle-loaded MSCs than in the case of injection of PTX-based MSCs or PTX-PLGA nanoparticles alone [190]. Another interesting study found that transactivation of transcription (TAT) functionalisation of Paclitaxel-loaded PLGA polymeric nanoparticles reduced the intracellular accumulation and retention of nanoparticles in laboratory-prepared mesenchymal MSCs in both primary tumours and metastases in a mouse orthotopic model of lung cancer. Moreover, the therapeutic use of nanoengineered MSCs increased MSC viability, inhibited cancerogenic growth and improved overall survival in Fox Chase SCID Beige mice compared to the use of the free or nanoparticle-encapsulated drug [191]. In summary, the data suggest that MSCs bearing nanoparticles may represent an effective potential vehicle for tumour-specific delivery of anti-cancer drugs, resulting in significantly improved therapeutic efficacy.

#### 3.2.4. MSCs Carrying Anti-Cancer MicroRNAs

Another set of molecules that may represent a potential strategy for the therapy of human cancers are microRNAs (miRs), due to their ability to modulate post-transcriptional gene expression. In vitro studies and animal models use MSCs that “carry” a variety of miRs packed into extracellular vesicles (MSC-EVs) and deliver them to neoplastic cells or neighbouring tumour niche cells to induce a therapeutic anti-cancer effect. MSC-derived exosomes may also be used as delivery vehicles to transfer genetic materials and biomolecules, such as mRNA, DNA and non-coding RNAs, as well as oncolytic adenoviruses (Ads), viral vectors and anti-neoplastic drugs homing to specific sites and recipient cells [52,192,193,194,195]. In one preclinical study, researchers used lentiviral vectors to construct ex vivo cultured bone marrow-derived MSCs as natural biofactories for exosomes carrying miR-124a, and these were applied against multiple patient-derived glioma stem cell (GSC) lines and in a male athymic nude mice (nu/nu) model [193]. Furthermore, in vitro therapeutic use of GSCs with exosomes containing miR-124a (Exo-miR124) led to a biologically significant reduction in GSC viability and clonogenicity compared to control systems, and the in vivo treatment of mice with intracranial GSC267 after systemic administration of Exo-miR124 resulted in half of the experimental GSC xenografts demonstrating long-term survival. Interestingly, it appears that miR-124a acted by silencing Forkhead box (FOX)A2, a known target of miR-124a, and that apoptotic cell death correlated with FOXA2-mediated aberrant intracellular lipid accumulation. A similar study examined the in vitro delivery of exogenous miR-124 to glioblastoma multiforme (GBM U87) cells by human umbilical cord Wharton’s jelly MSCs (WJ-MSCs) [194]. The WJ-MSCs were characterised by functionally significant exosome-dependent or -independent anti-cancer effects associated with decreased *CDK6* target gene luciferase activity; they also inhibited U87 cell proliferation and neoplastic cell migration and increased the chemosensitivity of GBM cells to Temozolomide (TMZ) in endometrial cancer treatment. These findings were confirmed in a later study showing that human umbilical cord mesenchymal stem cell (hUC-MSCs)-derived extracellular vesicles inhibited endometrial cancer (EC) cell proliferation and migration by delivering exogenous tumour suppressor miR-302a through a pronounced neoplastic-homing ability [195]. The researchers reported that the miR-302a levels were significantly decreased in EC cancer tissues compared to adjacent non-cancerous tissues. It is believed that the miR-302a overexpression in the cancer cells inhibited cell proliferation and migration, both of which were blocked in cancer cell culture with miR-302a-loaded extracellular vesicles derived from hUC-MSCs. Additionally, laboratory-modified miR-302-rich MSC-EVs significantly inhibited pro-tumorigenic cyclin D1 expression and suppressed the AKT signalling pathway in EC cancer cells in vitro. This suggests that exogenous miR-302a delivered by EVs has great potential as an effective anti-cancer therapy.

#### 3.2.5. Limitations of MSC-Based Therapies for Human Tumours

Unfortunately, MSC-based therapies for various human cancer diseases also have significant technical and biological demands. The successful engraftment of MSCs transduced with adenoviral vectors or MSCs “carrying” oncolytic adenoviruses or anti-cancer payloads and drugs, and their satisfactory survival, remains problematic. One potential solution is to use appropriate biomaterials, such as Gliadel, or a thermos-responsive biodegradable paste, preferably with their own anti-cancer or repair activity, as a scaffold to improve the retention of transplanted stem cells. Such frameworks have been found to demonstrate anti-cancer and repair effects in the tumour niche or in inflamed or damaged tissue in vitro and in vivo [196,197,198,199,200]. A number of studies have described new methods for delivering therapeutic MSCs to biomaterials for the treatment of specific human cancers and pathological conditions [201,202]. One such study examined the implantation of biodegradable fibrin scaffolds of seeded MSCs into a resection cavity after postoperative brain cancer surgery. The results confirmed the removal of residual tumour cells, which could be a cause of later local recurrence, as well as improved anti-cancer MSC persistence as well as longer cancer-free survival [201]. Another study proposed the use of an innovative immunotherapeutic organoid using human mesenchymal stromal cells (hMSCs) genetically modified to secrete bispecific anti-CD33-anti-CD3 antibody (bsAb); these cells were placed in a small biocompatible star-shaped poly(ethylene glycol)-heparin container. The organoid demonstrated the slow release of bispecific antibodies and enabled effective minimally invasive immunotherapy in acute myeloid leukaemia (AML). The macroporous biohybrid cryogel platform effectively increased the proliferation and survival of the MSCs, allowing them to release bsAb over a longer period of time in vitro and in vivo. Moreover, the experiment led to the sustained active release of bsAb, resulting in high levels capable of inducing a T cell-mediated anti-tumour response and rapid regression of CD33^+^ blasts in AML [202]. Interstitial implantation of alginate-encapsulated cell expressing a soluble form of leucine-rich repeat and immunoglobulin-like domain 1 (sLrig1) significantly inhibited the development and growth of patient-derived glioblastoma multiforme in a mouse orthotopic xenograft model [203]. The usage of Slrig1, a negative regulator of the oncogenic epidermal growth factor receptor (EGFR) family, disrupted downstream signalling in both wild-type and constitutively active EGFR mutated glioma cells (EGFRvIII) in vitro and in vivo. Interestingly, the other noticed effectors included MAP kinase but not AKT signalling. Hence, initial research suggests that mesenchymal stem cells may be promising therapeutic reference points. In the future, they may be used to treat various diseases, including cancer, due to their ability to inhabit damaged tissues and differentiate into various types of cells and their pleiotropic effect. However, their potential value in the treatment of human cancers is hampered by the fact that preclinical and clinical studies indicate they demonstrate both anti-cancer and pro-tumour effects, which constitutes important limitations for further research.

Despite these significant limitations, current analyses indicate that the latest MSC-based therapies offer considerable anti-tumour potential in human patients and may represent effective personalised anti-cancer therapies. Among the MSC-based therapies, the most promising challenge in developing effective minimally invasive cancer treatment is the use of MSCs as “Trojan horses” to deliver various therapeutic agents to the tumour niche or neoplastic cells. Another ongoing problem is that the interactions occurring between MSCs and cancer cells are relatively poorly understood, and further knowledge is needed of the activities occurring between them to improve their safety as therapeutic strategies. In this regard, the use of MSC-derived extracellular vesicles (MSC-EVs) as “cell-free carrier” therapy is becoming an increasingly promising option to remove or mitigate the risks associated with the use of live cells. It is worth noting that in practice, highly effective MSC-based therapies constitute an acceptable option for known anti-cancer therapeutic procedures, both as mesenchymal cells directly targeting the destruction of the neoplastic lesions and regulating tumour niche remodelling, and as a way of minimising the side effects of cancer treatments, such as chemoradiotherapy-induced xerostomia (NCT03874572), cisplatin-induced acute renal dysfunction (NCT01275612), cardiomyopathy (NCT02509156) or radiation-induced haemorrhagic cystitis (NCT0284864), etc. Despite continual progress and the growing body of research on using delivery MSCs, it is difficult to identify clear published clinical studies that could be directly translated into clinical outcomes, which unfortunately hinders further progress in the therapeutic application of MSC-based therapies. Nevertheless, the use of MSC implants is promising and seems to be a safe potential alternative to other therapeutic strategies. Finally, it should be noted that despite the limitations and engineering difficulties that researchers encounter, further in-depth research will hopefully eliminate the complications and symptoms associated with the use of MSC-based therapies for various origin human cancers. An increasing number of cell-free MSC therapy studies indicate that there is a real hope of generating a safe and effective therapeutic product that can inhibit or kill tumour cells, thus improving the survival and quality of life of patients in the advanced stages of this devastating and irreversible disease [52,204].

Most intensive and wide-ranging studies on the influence of MSCs on cancer development and growth, their immunomodulatory abilities and potential anti-cancer therapies are based on in vitro and in vivo 2D human cancer co-culture systems, most of which have employed isolated BM-derived and adipose tissue-derived MSCs. It should be noted, however, that in the last few years, new approaches using more complex 3D in vitro models have also been proposed. Although preclinical 3D dynamic culture systems are still under initial development, they more closely mimic the features of the tumour microenvironment in vivo and interest in them is constantly growing [204,205]. A summary of the various available preclinical studies with 3D models, as well as their limitations, is presented in more detail in a recent publication by Avnet et al. [205].

Another important problem may be the immunogenic phenotype of MSCs, which may have a double-edged effect if the tested cellular systems are not properly controlled in vitro and in vivo. It is believed that resident MSCs can acquire immunosuppressive properties upon exposure to elevated levels of pro-inflammatory cytokines and then provide support for tissue repair and inhibit tumour initiation and growth through the secretion of TME components. Unfortunately, in the presence of cytokines, i.e., IFN-γ. IL-6, IL-8, SDF-1α and TNF-α, or tumour-derived agents, i.e., IL-10, IDO, and TGF-β, and in response to signals generated by direct contact of cancer cells and MSCs, mesenchymal stromal cells may adopt an immunosuppressive phenotype that affects both innate cells and development of adaptive mechanisms and facilitates the tumour’s escape from immune surveillance. Moreover, the anti-tumour phenotype of MSCs may also facilitate neoplastic progression via the production of chemokines, such as CXCL 1/5/6 /7/8, and CCL5, and growth factors, such as VEGF, EGF and PDGF; this can result in the acquisition of key genetic and epigenetic changes by the cancer cells, which may protect them against cytotoxic cells and drugs and promote metastasis. However, it is important that, as noted above, genetically manipulated MSCs can also have significant anti-cancer effects by delivering and expressing various anti-cancer agents, including type I interferon (IFN-α and IFN-β), CXCL1, IL-2, IL-12, cytokine deaminase, oncolytic virus, TRAIL and nanoparticles, thus reducing the risk of treatment failure. Another important problem is the natural heterogeneity of MSCs, which requires researchers to precisely identify subpopulations of MSCs in the tumour milieu and determine their mutual relationships to other immune cells and TME compounds. It also remains unsatisfactory to determine whether the derived MSC types are stable or transient under experimental conditions and whether one subpopulation can transform into another in response to various microenvironmental stimuli; it is also unclear what proportion of the stromal response to injury or cancer is directed by MSCs rather than other more differentiated stromal cells [52,204].

Also, it must be considered that autologous or allogeneic MSCs actively interact with components of tumour microenvironment, and despite several studies, the therapeutic benefits of autologous or allogeneic MSCs remain inconclusive. This may also indicate that the functional plasticity and heterogeneity typical of MSCs caused by different donors and MSC subpopulations of different origin, as well as interactions with cells present in the niche, may be the reason for the poor results observed in some studies. Allogeneic and autologous MSCs offer many important advantages, e.g., donor selection, unrestrictive cell dose, hypo-immunogenicity, immediate availability, cost effectiveness, better safety and greater suitability for immunocompromised patients. Unfortunately, both MSC selection methods have significant drawbacks: allogeneic transplantation is associated with potential immune rejection, donor–donor heterogeneity, specific immunological memory and quick clearance after infusion, while autologous MSCs are characterised by long-time availability, limited cell dose, donor variability issues and prohibitive cost. However, on balance, despite their own merits and limitations, the allogeneic approach seems to be superior [206,207,208].

Therefore, to optimise anti-cancer treatment, it is necessary to thoroughly understand the mechanisms of action of MSCs, determine their role in new therapies using pleiotropic and multifunctional cells, and define new biomarkers enabling the prediction of the response to therapy; these will be of key importance in developing alternative treatment regimens and allowing accurate patient stratification. New research into the use of alternative types of MSCs to deliver therapeutic agents to the tumour niche or neoplastic cells will open up new fields in cancer treatment, including HNC, although the clinical benefits remain unclear.

The current and new discoveries in the field of MSC-based therapy, and potential therapeutic targets in human cancers of various origin, are summarised in Figure 2.

### 3.3. The Pharmacological Strategies of MSC-Based Treatment for HNC

Most patients with squamous cell head and neck carcinomas (HNSCCs) are in an advanced stage (stage III/IV) due to late detection and are treated with radiochemo-therapy in this region. This type of complex anti-cancer therapy is frequently associated with failures, resulting in shortened 5-year and overall survival, cancer-free survival and local recurrences and distant metastases, as well as problems with speech and swallowing, oral infection and dental caries. The latter significantly reduces quality of life and can give rise to complications such as salivary gland hypofunction (MARSH) and xerostomia (RIX), subjectively named dry mouth or Sjögren syndrome, following radiotherapy [164,209]. Despite the use of increasingly more precise and effective radiological methods, i.e., intensity-modulated radiation therapy (IMRT), most patients with HNC experience dose-dependent damage and degeneration of salivary gland parenchyma cells and decreased saliva production [164,209,210,211]. To identify alternative treatments, several studies recorded on the ClinicalTrials.gov website (data from ClinicalTrials.gov., U.S. National Library of Medicine, available on 30 April 2024) have evaluated the use of easily accessible multipotent and pleiotropic adult progenitor mesenchymal cells in anti-neoplastic therapy and in the preventive treatment of xerostomia. Importantly, most of these studies focus on bone marrow (BM-MSCs) and stromal cells isolated from adipose tissue (A-MSCs), which demonstrate high therapeutic potential due to their specific anti-tumorigenic features, i.e., trophic properties, anti-inflammatory, anti-apoptotic activity, pro-angiogenic characteristics and immunomodulatory effects. These properties make the cells a very promising direction of research for use in anti-cancer therapy and as a source of cells that can promote the regeneration of the salivary gland parenchyma and restore radiation-induced damage. Unfortunately, despite progress in various preclinical in vivo models and human studies, only a few publications have examined the use of MSC transplantation as a potentially curative treatment option in HNC [164,165,166,167,168,169,170,171,172,173,174,175,176].

#### 3.3.1. The Use of MSC-Based Treatment for HNC

Most importantly, it should be emphasised that there is currently only a single MSC-application-based research study has evaluated the safety and effectiveness of genetically engineered mesenchymal stem cell therapy as a complement to standard treatment in patients with head and neck cancer (NCT02079324) [52]. A phase I clinical study (NCT02079324) by a team of Korean scientists evaluated the maximum tolerable dose, safety and efficacy of intratumorally injected GX-051 therapeutic agent, a genetically modified mesenchymal stem cell treatment, against HNC. The clinical study started in 2014 and had only a few secondary outcomes: anti-tumour response by Response Evaluation Criteria in Solid Tumours (RECIST 1.1) on computed tomography, changes in INF-γ and IL-12 levels in blood by ELISA compared to baseline after GX-051 intratumoural injection, assay of immune cells (inter alia CD4^+^ T cell, CD8^+^ T cell, NK cell) by FACS on day 1 (baseline), day 29 (end of treatment) and day 57 (follow up); safety profile was examined by vital signs, physical examination, clinical laboratory tests, and CT of NHC region (data from ClinicalTrials.gov., U.S. National Library of Medicine, available on 30 April 2024). Although the research was planned to be completed in 2015, no results have yet been posted [52].

#### 3.3.2. The Use of MSC-Based Treatment for Irradiation-Induced Salivary Dysfunction (MARSH) and Xerostomia (RIX)

It should be noted that all the existing prospective, randomised and controlled preclinical in vitro and animal models and clinical in vivo or human tissue studies evaluate the use of MSCs derived from various origins for treating irradiation-induced salivary dysfunction (MARSH) and xerostomia (RIX) in specific head and neck cancers [164,165,166,167,168,169,170,171,172,173,174,175,176]. Many of these are reviewed in the present paper, particularly those concerning radiotherapy-induced xerostomia, e.g., the PROSPERO study (www.crd.ac.uk/prospero; accesed on 30 April 2024), registration number CRD42021227336, and I/II phase clinical studies such as NCT04489732, NCT047765392, and NCT03874572 [164,165,166,167,168,169,170,171,172,173,174,175,176]. However, as an analysis of all the available publications exceeds the scope of this study, only the most interesting and promising articles were included, based on data obtained from ClinicalTrials.gov, U.S. National Library of Medicine (available on 30 April 2024) and PubMed/Medline/EMBASE Library database and Cochrane Database of Systematic Reviews.

One of the interesting most recent qualitative meta-analyses was study number CRD42021227336 on PROSPERO by Carlander et al. [164]. It examined the effect of mesenchymal stromal/stem cell therapy on the salivary flow rate (SFR) in experimentally radiation-induced salivary gland hypofunction in preclinical interventional in vivo models. The analysis included a total of 16 in vivo preclinical studies in animal models and 13 meta-analyses conducted in animal experiments. The researchers used MSCs derived from bone marrow (BM-MSCs), adipose tissue (A-MSCs) and salivary gland tissue (SG-MSCs), which were administered intravenously, intraglandularly or subcutaneously. The summary results of the included studies indicated that MSC-based therapy significantly increased the SFR. Furthermore, the results of the preclinical in vivo indicated that treatment restored salivary gland functionality and regenerated tissues following radiotherapy in acinar tissue, vascular areas, and paracrine functioning without reported serious adverse events. Additionally, the most noticeable effect on the SFR was observed when MSCs were provided through intraglandular administration compared to systemic transplantation. The meta-analysis also discussed the importance of the time from radiation to administration of MSCs, which was directly related to the number of apoptotic cells [212,213,214,215]. The authors unanimously emphasised that transplantation of isolated tissue-specific human stem cells to radiation-damaged salivary glands rescued hyposalivation, restored acinar and duct cell structure, and it decreased the amount of apoptotic cells. These observations indicated that MSC-based intraglandular therapy could be protective, especially in the acute period of radiological treatment [212,213,214,215]. The researchers also discussed two important studies that assessed the effect of intravenously administered MSC therapy on the structure and function of salivary glands after high-dose radiation therapy in head and neck cancer [215,216]. The studies presented similar final conclusions, indicating that transplantation of MSCs by intravenous infusion in animal models immediately after local radio-irritation significantly increased the salivary gland weights and improved SFR and acinic cell function; it also resulted in higher amylase production and micro-vessel densities in MSC-treated salivary glands than in those that only received irradiation. Moreover, in the examined groups of experimental animals, systemic transplantation of mesenchymal cells resulted in increased colonisation of the salivary glands, which was associated with protection against irradiation-induced cell loss and induction of transdifferentiation into glandular cells. Similar conclusions from the meta-analysis protocol were presented by Jansson et al. [165], who conducted PROSPERO project CRD42021227336, based on data from MEDLINE/PubMed and Embase Library databases and validated according to the peer review of electronic search strategies (PRESS). An objective systematic review including animal intervention studies confirmed the efficacy and safety of MSC implantations in treating post-radiation salivary gland function disturbances and xerostomia. Moreover, multipotent adult MSC cell therapy based on intra-glandular MSC implantation and intravenous and/or intramuscular injections was shown to enhance the unstimulated SFR, achieving positive changes in the salivary gland morphology, cytoprotection, apoptosis and organ vascularity. Importantly, however, the authors of these meta-analyses emphasised the high heterogeneity among the included studies with regard to the MSC derivation, species, strains, age, radiation-therapy dose, dosage protocols, administration route of MSC therapy, frequency of treatment and time between radiation and first treatment, as well as frequent insufficient group size; this variation can result in significant limitations. Moreover, it was impossible to analyse the relationship between the radiotherapy duration and the effectiveness of MSC therapy due to insufficient data. These limitations should always be taken into account when translating results and conclusions to a clinical setting.

Moreover, and importantly, several in vivo preclinical studies and randomised placebo-controlled phase I/II clinical trials have analysed the safety and efficacy of autologous MSCs treatment for radiation-induced xerostomia (RIX) and restoration of salivary hypofunction (MARSH) in humans, including NCT04489732, NCT047765392 and NCT03874572. The MARSH and RIX symptoms are frequently observed in human cancer patients, and this can be a prognostic factor that can also determine the quality of life in head and neck carcinoma. For example, two studies relating to the use of MSCs as a therapeutic option in the restoration of salivary hypofunction in HNC in humans were recently published by Blitzer et al. [171,173,174]; they were conducted under an approved Food and Drug Administration Investigational New Drug application using an institutional review board-approved protocol (NCT04489732). The study was performed as a pilot, first-in-human study. It described the clinical and morphological effects of a single injection of autologous bone marrow-derived mesenchymal stromal cells (BM-MSCs) into the right submandibular gland stimulated with INF-γ for the treatment of radiation-induced xerostomia [173]. The proposed therapy was found to be safe and well tolerated, and the endpoints analysis showed a clear tendency towards increased saliva production, which persisted in 50% of patients both one and three months after MSC injection; this was accompanied by improved quality of life indicators. The same group of researchers used the same innovative approach to treat RIX and MARSH and improve the quality of life in a phase I dose-escalation trial of patients with xerostomia and salivary hypofunction after radiation therapy for head and neck cancer [174]. This analysis, registered as NCT04489732 by the World Health Organization International Clinical Trials Registry Platform, examined the effect of injecting the pro-growth secretome of IFN-γ-stimulated marrow-derived autologous stromal cells (BM-MSCs) taken from patients with HNC who had undergone radiation or chemoradiotherapy into the submandibular gland. A total of 21 to 30 subjects (9 to 18 in phase I study, 12 in expansion cohort) were included in the study group. The study included two endpoints: the dose-limiting toxicities occurring within one month of the submandibular region BM-MSC injection. The outcomes of the saliva amounts and composition, ultrasound images of the salivary glands, and the quality of life from 3 to 24 months after treatment were recorded. The researchers indicated that such autotransplantation of BM-MSCs stimulated with IFN-γ into the salivary glands after radiotherapy or chemoradiotherapy could be an innovative method of treating RIX and MARSH, which also translates into restoring a better quality of life.

A recent study by Blitzer et al. [171] analysed the functionality of bone marrow mesenchymal stromal cells (BM-MSCs) derived from HNC patients in an FDA-IND enabling study regarding MSC-based treatments for RIX. In this pilot clinical study, bone marrow aspiration was performed in HNC patients who had completed radiotherapy two or more years earlier; the aim was to isolate and culture MSCs after IFN-γ stimulation. The MSCs were additionally implanted in mice with radiation-induced xerostomia and changes in the salivary gland histology and saliva production were examined. The results of this preliminary study clearly indicated that autotherapy with IFN-γ-stimulated MSCs in HNC patients led to the acquisition of an immunosuppressive mesenchymal stem cell phenotype and higher protein expression, i.e., GDNF, WNT1, and R-spondin 1 as well as pro-angiogenesis and immunomodulatory cytokine activity. Moreover, in a mouse model, MB-MSC injection after radiation down-regulated the loss of acinar cells, decreased the formation of fibrosis, and increased salivary production. Another noteworthy study was the analysis performed by Grønhøj et al. [172], who examined the safety and efficacy of MSC-based therapy for radiation-induced xerostomia in a randomised, placebo-controlled phase I/II trial (MESRIX). This interesting randomised trial included 30 patients who had previously received radiotherapy for a T1-2/N0-2A, human papillomavirus-positive (HPV^+^) oropharyngeal squamous cell carcinoma (OPSCC) and who underwent autologous adipose tissue-derived mesenchymal stem cell (A-MSCs) therapy. In this analysis, the investigators also noted that the use of A-MSC therapy was substantially effective for radiation-induced hypofunction and xerostomia, and it led to significantly improved salivary gland secretion, increased whole salivary flow rates (SFRs), reduced RIX and MARSH symptoms and improved patient-reported outcomes for months compared to the placebo arm. Interestingly, core-needle biopsies conducted in the trial confirmed the up-regulation in serous gland tissue and decreases in adipose and connective tissues in the ASC-arm compared to the placebo-arm; however, no differences between groups in terms of the gland size or intensity were observed. Importantly, the authors did not observe any adverse events of this therapeutic procedure.

Also very interesting from a clinical point of view are three recent publications on phase I/II randomised trails (MESRIX-I and MESRIX-II) in patients with radiation-induced xerostomia by Lynggaard et al. [169,170]. The first is an investigator-initiated, randomised, single-centre, placebo-controlled trial (MESRIX-I; EudraCT number: 2014-004349-29) involving the injection of allogenic A-MSCs or placebo into both submandibular glands in patients with oropharyngeal squamous cell carcinoma (OPSCC), followed by radiotherapy for a minimum of two years. The study’s primary endpoint was the observation of serious adverse events after MSC-based treatment. The secondary endpoint was the presence of an entire SFR and local symptoms associated with RIX. The results indicated that during follow-up, no side effects appeared to be related to the MSC treatment. Moreover, the authors observed that the whole saliva flow rate increased and OPSCC patient-reported xerostomia symptoms decreased in the patients [170]. In the second study, Lynggaard et al. [169] examined the effectiveness of intraglandular off-the-shelf allogeneic adipose tissue derived mesenchymal stem cell (A-MSCs) therapy in HNC individuals with salivary gland hypofunction (MESRIX-II; EudraCT number: 2018-003856-19). In this safety study, the occurrence of adverse events and the unstimulated and stimulated saliva SFR indicators and composition of saliva were assessed based on the results obtained from the EORTC QLQ-H&N35 and the XQ xerostomia questionnaires, as well as on blood samples and salivary gland scintigraphy. The authors noted no treatment-related serious adverse events during a four-month observation period; they also reported a relevant increase in the saliva flow rates, a reduction in the feeling of dry mouth and improved swallowing based on patient-related data. These results were presented in another phase I trial using allogeneic MSCs for radiation-induced hyposalivation and xerostomia (MESRIX-SAFE) by the same authors; the study is registered as NCT03874572 according to ClinicalTrials.gov. (U.S. National Library of Medicine, available on 30 April 2024). This open-label clinical study confirmed the safety and feasibility of local submandibular gland therapy with the use of the autologous adipose-derived mesenchymal stem cells from healthy donors in previous oropharynx cancer patients. A non-randomised, open-label, phase I exploratory study by Strojan et al. also examined the positive impact of MSC-based therapy as a potential medical procedure in patients with head and neck tumours [167]. The researchers presented a trial clinical protocol used to evaluate the safety and preliminary efficacy of allogeneic MSCs derived from human umbilical cord tissue (hUC-MSCs). The study itself assessed the clinical effect of hUC-MSC implantation under ultrasound guidance into both parotid glands and both submandibular glands. It included 10 oropharyngeal cancer patients with post-radiation xerostomia and no evidence of disease recurrence during two or more years after chemoradiation (intervention group) and 10 healthy volunteers (control group). The initial data, obtained four months post-procedure, indicated that allogeneic hUC-MSC treatment yielded positive and expected effects regarding the salivary flow and composition, scintigraphic evaluation of MSC grafting, retention and migration, and positive subjective survey data regarding xerostomia and quality of life in the OPSCC patients. The authors confirmed that this is a potential and innovative approach for the treatment of RIX syndrome, not only due to the improvement of the functional and morphological features of the salivary tissue but also to its non-invasive collection procedure, flexibility of cryobanking and biological advantages.

Following on from of the MESRIX studies above, Jakobsen et al. [168] presented a single-centre, double-blinded, randomised, placebo-controlled, phase II study aimed at investigating the safety and efficacy of mesenchymal stem cell for MARSH and RIX in previous head and neck cancer patients (MESRIX-III), which was registered as NCT04776538 at the ClinicalTrials.gov (U.S. National Library of Medicine, available on 30 April 2024). The study began in March 2021 and was completed in September 2023. It included 120 participants who had previously been treated with radiotherapy. The participants were divided into two groups of equal numbers (60:60 in HNC individuals vs. placebo) and received ultrasound-guided injection of either allogeneic adipose-derived mesenchymal stem cell (A-MSCs) or placebo into the submandibular glands. The primary endpoint was the observation of the unstimulated whole saliva flow rate (SFR). The secondary endpoints included the change in the flow rate of stimulated whole saliva, quality of life based on EORTC QLQ Module for H&N-35 and XQ questionnaires, saliva composition and immune response determined in blood samples (human leukocyte antigen—HLA response) to stem cells, before treatment and at the four- and twelve-month follow-up. Importantly, the final results and conclusions have already been presented [166]. The analysis of the obtained phase II placebo-controlled trial data clearly indicated that the use of allogeneic A-MSC local submandibular gland therapy resulted in a statistically significant SFR up-regulation, a decrease in sticky saliva, fewer notable swallowing difficulties and radiation-induced xerostomia symptoms compared with pretreatment baseline and placebo. Hence, injecting A-MSCs into the submandibular salivary gland may be a promising therapeutic procedure with high effectiveness and safety and may constitute a potential new method of treating hyposalivation and salivary gland regeneration in patients after radiotherapy [166,168].

Other studies have also analysed the possible mechanisms of action of MSCs on salivary gland tissue, favouring higher proliferative activity, greater tissue remodelling, higher density of acinar SG cells and lower post-radiation inflammation and fibrosis [164,217,218,219,220]. In vitro analysis data confirms that the use of intraglandular MSC transplantation increases the expression of epithelial markers (KRT7 and KRT18) and genes related to organ structure (SMR3A, AMY2A5, PRB1, AMY1, CLDN22, PRPMP, AMY1A, AQP5, α-SMA and CD31), as well as the genes encoding proteins involved in cell migration, survival and differentiation (SDF1-CXCR4 and Bcl-2) [217,218,219,220]. It also appears that a variety of growth factors and other active biomolecules, including VEGF, HGF, COX-2, BDNF, GDNF, EGF, IGF1, NGF, FGF10 and MMP-2, are responsible for angiogenesis and paracrine function, and for promoting tissue repair and restoring the salivary glands [164]. A study on the consequences of intraglandular allogeneic AT-MSC-based therapy and induced changes in the salivary proteome of irradiated HNC patients was performed by Lynggaard et al. [175]. The results indicate the presence of significant differences in over a hundred human proteins associated with post-transplantation tissue regeneration between the saliva of the HNC patients with radiation-induced hypofunction and that of healthy controls. Moreover, the authors noted an increase in the regenerative effects on the salivary proteome; however, this proteome did not return to a healthy state when compared to the control individuals.

In conclusion, numerous studies clearly indicate that MSC transplantation offers various significant benefits in head and neck cancers, leading to better regenerative outcomes by contributing to various aspects of salivary gland tissue repair, including their cell proliferation, migration, angiogenesis and weight gain. Unfortunately, it should also be emphasised that large-scale standardised research and further extensive systematic reviews and meta-analyses and meticulous studies on in vitro and animal models, as well as comprehensive, randomised, well-planned clinical trials in humans, are necessary to definitively determine the clinical value and effectiveness of MSC-based therapy methods.

#### 3.3.3. Limitations of MSC-Based Treatment for HNC

However, it is also necessary to mention the important limitations of these analyses. Although numerous preclinical in vitro and in vivo studies based on various animal models have been carried out to date and promising results have been obtained, it is difficult to indicate clear and conclusive findings regarding the role of MSCs in the stages of carcinogenesis and tumour progression in HNC. The majority of recent research indicates that MSCs have the ability to modulate phenomena directly and indirectly related to cancer occurring in the tumour microenvironment, i.e., apoptosis, differentiation, proliferation, invasion and metastasis, angiogenesis and growth factor signalling in HNC, as well as other cancers. Unfortunately, the ability of active MSCs of various tissue sources to regulate the initiation of cancer or modulate its course often differs from the preclinical data obtained in experimental models. This may be due to many reasons. Firstly, there may not have been any methodological standardisation of the mesenchymal stem cell activity assessment, translational research model or analytical protocol, or laboratory method. Secondly, many preclinical studies do not take into account certain factors that may have a significant impact on the pro- or anti-tumour role of MSCs, i.e., the use of cells from different regions of the head and neck and different types and origins of tumours. It is also important that some signalling pathways through which MSCs can interact and determine the function of various cells, such as other populations of stromal cells, stem cancer cells and cells of the innate and adaptive immune system, in the cancer microenvironment are not yet fully known and understood. Hence, the preclinical researcher cannot accurately predict the structural and functional differences in metabolic competences and pathways/biomarkers related to the stages of carcinogenesis modulated by MSCs with respect to the clinical analyses and final clinical conclusions.

It is important to emphasise that currently, unfortunately, only symptomatic treatment is available for patients suffering from radiation-induced xerostomia and hypofunction and hyposalivation of salivary gland. Therefore, clinical trials using mesenchymal stromal/stem cells from various sources in modern potential MARSH and RIX treatment strategies in cancer patients are currently of great interest. Existing data from preclinical in vivo studies, systematic reviews, meta-analyses and I/II phase clinical trials clearly confirm that MSC therapy has a significant impact on the restoration of salivary gland function and tissue regeneration after radiotherapy. Moreover, no serious side effects have been observed so far, and intraglandular transplantation was associated with a significantly better therapeutic effect than systemic transplantation. Therefore, in summary, MSC-based therapy shows significant therapeutic potential in the treatment of xerostomia and radiotherapy-induced hyposalivation, but large-scale, sufficiently large, randomised human trials are needed to confirm its effectiveness in clinical settings.

## 4. Conclusions

Previous findings indicate that mesenchymal/stem cells play a dual role in the progression of HNC, being able to promote the development, progression and aggressiveness of the tumour by exerting differential effects on different phases of carcinogenesis. MSC activity may influence key intracellular signalling pathways, i.e., NF-κB, ERK, MAPKp38, and JNK kinases, and modulate the PI3K/AKT pathway and cell cycle progression at crucial transition points. These intracellular pathways can also reduce the effectiveness of chemotherapy against HNC by influencing cell proliferation and angiogenesis upon stimulation by MSCs. Moreover, MSCS may also act as anti-inflammatory stimulators, thus influencing immunity, oncogenic cellular signalling and apoptosis inhibition, and by regulating the cell cycle and angiogenesis, enhancing the carcinogenic phenomenon of HNC.

Further translational research is needed to expand the current knowledge of MSCs and understand their role in the molecular pathways associated with cancer development, progression, and metastasis, as well as those leading to tumour resistance; research should also assess the use of state-of-the-art comprehensive chemoradiotherapy in human cancers, including HNC. Indeed, numerous in vitro and in vivo studies clearly demonstrate that combining classical chemotherapeutics with stemness modulators can yield clinical benefits. Furthermore, recent evidence indicates that MSCs may be considered a potential and promising method for the delivery of anti-neoplastic agents to the tumour milieu or as therapeutic onco-modifiers of genes, or as vehicles for transporting chemotherapeutic drugs. This therapeutic approach, targeted at proliferating chemotherapy-resistant cancer cells and stem cells, may better inhibit the progression and recurrence of cancer.

In conclusion, further multicentre studies are needed to identify more effective and less toxic MSC-based strategies aimed at preventing tumour escape from immune surveillance and which can promote the development of new treatments for HNC. Importantly, new research is necessary to understand the phenomena promoted by MSCs, translate them into clinical conclusions, and ultimately propose new therapeutic combinations and develop biomarkers for personalised therapy.

## Figures and Tables

**Figure 1 cells-13-01270-f001:**
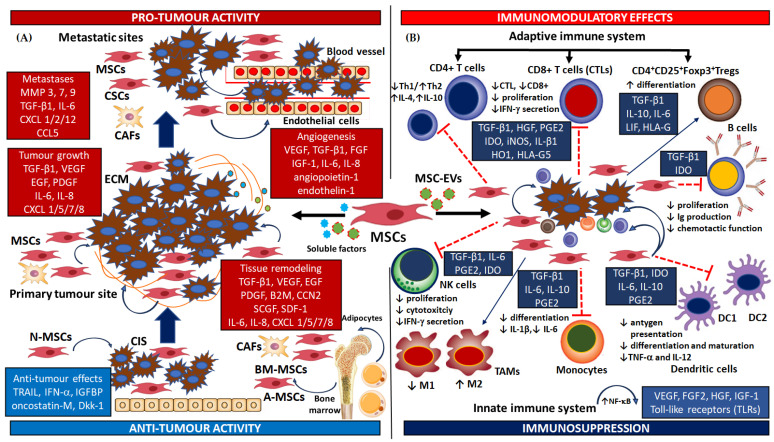
The schema of dual roles of MSCs in TME and their pro- and anti-tumorigenic activity. (**A**) Dual effect of tumour-associated mesenchymal stromal cells (MSCs) in head and neck cancer (HNC). MSCs can affect both tumour progression or tumorigenesis inhibition, with a tendency to promote the former. MSCs in the TME show pro-tumorigenic activity on neoplastic cells directly via cell-to-cell contact or extracellular vesicles (MSC-EVs). MSC-EVs include key immune-modulating agents, proangiogenic factors, pro-survival biological agents or soluble factors stimulating cell mobility and extracellular matrix modulators, which favour higher tumour aggressiveness. MSCs isolated from squamous cell carcinomas can mediate cancer progression by secreting pro-inflammatory and pro-angiogenic cytokines such as IL-6 and IL8, which also promotes the recruitment of TAMs, CAFs, T cells, neutrophils and other MSCs. In addition, TGF-β1,VEGF, EGF, PDGF and cytokines IL-6 and IL-8, as well as B2M, CCN2, SCGF, SDF-1 and chemokines such as CXCL 1/5/7/8 are all primarily responsible for tissue remodelling and growth. The formation of new vessels in the tumour niche, enabling the initial growth of neoplastic lesions and then tumour metastasis, is associated with the action of VEGF, IGF-1, TGF-β1, FGF, angiopoetin-1, endothelin-1, and cytokines IL-6 and IL-8, which facilitate further tumour progression and growth by enhancing angiogenesis, increasing vascularisation and the local/regional and general spread of tumour cells. Most of these MSC-derived factors directly regulate the epithelial–mesenchymal plasticity, proliferation, invasion and migration and even drug resistance in HNSCC cells. MSCs can also activate other key signalling pathways, soluble agents and modulators, having an opposite effect on tumour progression and aggressiveness. The anti-tumour MSC-related paracrine soluble secreted factors and MSC-derived molecules released from exosomes that inhibit tumorigenesis include anti-proliferative factors such as Dkk-1, oncostatin-M, a soluble Wnt antagonist, PTEN, BMP, cytotoxic agents such as TRAIL, IFN-α, IGFBP and TNF-α, anti-angiogenic factors or immunomodulatory agents. MSCs may also actively inhibit further tumour development by blocking cancer-related signalling, such as the PI3K/AKT, Wnt/β-catenin, and JAK/STAT pathways. (**B**) The immunomodulation mechanisms of MSCs and their role in immune cell activity. Adaptive immune system: the active MSCs in the tumour milieu inhibit the adaptive immune response through the secretion of mediators contained in exosomes (MSCs-EV) and soluble factors, such as IDO, TGFβ1, TNF-α, IFN-γ, PGE2, NO, HLA-G, HGF, IL-1β, IL-1α, IL-4 and IL-6; they also interact with various immune cell types, including T cells, B cells, DC cells, NK cells, monocytes and TAMs. This MSC activity constrains dendritic cell maturation, reduces T cell proliferation, enhances macrophage activation and polarises them from M1 towards M2; it also facilitates neutrophil mobility and affects the regulation of NK cells and invariant natural killer T (iNKT) cells. In addition, it can shift the balance of T cell differentiation from the Th1 to an anti-inflammatory Th2 phenotype and enhance the maturation of T helper cells into the CD4^+^CD25^+^Foxp3^+^Treg pathways, which can inhibit effector T cell responses and thus reduce anti-tumour immunity. The innate immune system: in tumorigenic tissues, local factors, such as the cytokine milieu TNF-α and endotoxin LPS, hypoxia and Toll-like receptor (TLRs) ligands, stimulate MSCs, promoting the large-scale secretion of growth factors such as VEGF, FGF2, IGF-1, or HGF by an NFκB-dependent mechanism, with the effect of driving tissue regeneration, angiogenesis, reducing anti-tumour immunity and allowing the effective escape of the tumour from immune surveillance. Abbreviations: MSCs: mesenchymal stromal cells; CSCs: cancer stem cells; BM-MSCs: bone-marrow-derived MSCs; A-MSCs: adipose tissue-derived MSCs; N-MSC: naïve MSCs; CAFs: cancer-associated fibroblasts; ECM: extracellular matrix; CIS: carcinoma in situ; MMPs (MT-MMPs): matrix metalloproteinases, also known as matrix metallopeptidases; TGF-β1: transforming growth factor beta 1; CXCL1/2/12: C-X-C motif chemokine ligand 1/2/12; CCL5: C-C motif chemokine ligand 5; VEGF: vascular endothelial growth factor; EGF: epidermal growth factor; PDGF: platelet-derived growth factor; FGF: fibroblast growth factors; TRAIL: TNF-related apoptosis-inducing ligand; INF-α: type-I interferon alpha; IGFBP: insulin-like growth factor-binding protein; Dkk-1: Dickkopf-related protein 1; HGF: hepatocyte growth factor; PTEN: phosphatidylinositol 3,4,5-trisphosphate 3-phosphatase; BMP: bone morphogenetic protein; TRAIL: TNF-related apoptosis-inducing ligand; PGE2: prostaglandin E2; B2M: beta 2 microglobulin; CCN2: cellular communication network factor 2; SCGF: stromal cell growth factor-beta; SDF-1: stromal cell-derived factor 1; IDO: indoleamine 2,3-dioxygenase; iNOS: nitric oxide synthases; HO1: heme oxygenase 1; CTLs: cytotoxic T cells; CD4^+^CD25^+^Foxp3^+^T_reg_: regulatory T cells, known as suppressor T cells; TAMs M1/2: tumour-associated macrophages M1/2, CD56^dim^/CD16^bright^ NK cells: activated natural killer cells, TLRs: Toll-like receptors; → activation mechanisms; ¦ inhibitory mechanisms; ↑: increase in expression and activity, ↓: decrease in expression and activity.

**Figure 2 cells-13-01270-f002:**
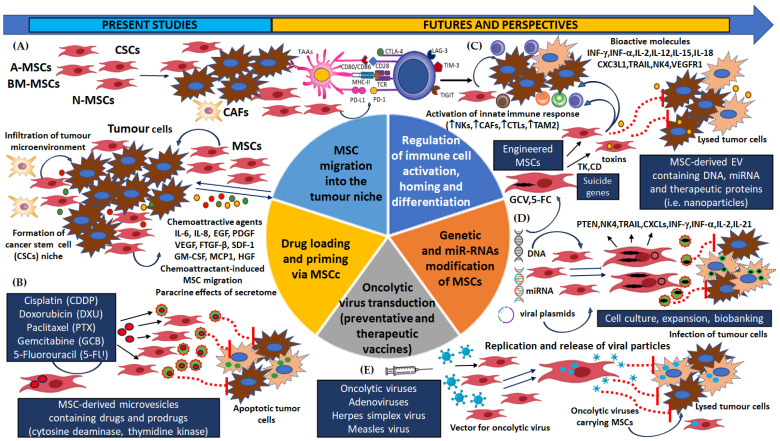
The current and new discoveries in the field of MSC-based therapy, and potential therapeutic targets in human cancers of various origin. Current studies and potential targets for MSC-based therapies of human cancer of various origins. MSCs and tumour cells may interact, thus influencing the MSC-based approaches for preventing tumour initiation and growth. (**A**) The chemotactic movement of MSCs towards a tumour environment and chemoattractant-induced infiltration of tumour niche by naive MSCs (N-MSCs), and adipose- or bone marrow-derived mesenchymal cells (A-MSCS, BM-MSCs); in addition, the paracrine effects of MSCs’ secretome and stimulation of activation cancer stem cells (CSCs) is driven by soluble factors such as growth factors and chemokines/cytokines, i.e., VEGF, PDGF, IL-8, IL-6, EGF or FGF2, SDF-1, G-CSF, GM-CSF, MCP1, HGF, and TGF-β. (**B**) MSCs have the ability to deliver therapeutic drugs such as Doxorubicin (DOX), Paclitaxel (PTX), Gemcitabine (GCB), 5-Fluorouracil (5-FU) and cisplatin (CDDP) directly to the tumour microenvironment. In addition to the direct use of MSCs in drug loading and priming, the secreted extracellular vesicles (MSC-EVs) isolated from MSCs represent an alternative approach to delivery. (**C**) MSCs also have a high regulatory potential regarding the role of immunocompetent cells present in the tumour niche. MSCs, by secreting or provoking the synthesis of bioactive molecules and immune-modulating agents such as INF-γ, INF-α, IL-2, IL-12, IL-15, IL-18, CXC3L1, TRAIL, NK4, and VEGFR1, may promote the regulation of immune cell activation/homing and immune cell differentiation and increase the activation of anti-tumorigenic innate immune response, among others by promoting the anti-cancer activity of cells NK, CAFs, CTLs, TAM2 immune cells. (**D**) Genetically modified MSCs containing inter alia suicide genes, DNA, miRNA, non-coding RNAs and other therapeutic proteins, such as growth factors, transcription factors, cytokines or viral plasmids, or nano-engineered MSCs carrying nanoparticles, can also be used to deliver a range of tumour-suppressing cargos directly into the neoplastic milieu. These cargos include tumour suppressors, i.e., PTEN, NK4, TRAIL, CXCLs, INF-γ, INF-α, IL-2, and IL-21 which can inhibit the formation of tumour cell culture, increase apoptosis and promote neoplastic expansion. They are also be used in cell biobanking. (**E**) MSCs can also be used in oncolytic virus transduction in constructed preventive and therapeutic cancer vaccines. MSC-cloaked oncolytic viruses such as adenoviruses, herpes simplex viruses, measles viruses, myxoma viruses hold great potential as a novel virus-delivery platform for the therapy of various cancers. Replication of viral particles in MSCs and subsequent release of them not only promotes apoptosis and lysis of cancer cells but may also be a potential effective engineered method of activating anti-cancer immune cells, i.e., cytotoxic T cells (CTLs), activated NK cells and tumour-associated macrophages (TAMs M2). MSCs also have great potential for possible advanced cell therapies in situ, e.g., in the pre-cancerous stage, or as an adjuvant therapy to reduce toxicity associated with systemic treatment, to prevent tumour recurrence after surgery, to reduce the effects of radiotherapy, etc. Numerous studies show that nanotechnology MSCs or virus-carrying MSCs in the neoplastic lesions can serve as an effective vehicle for tumour-specific drug delivery; they can also significantly improve the anti-tumour efficacy of conventional chemotherapy and promote tumour lysis through the activity of oncolytic viruses. However, the MSCs recruited in the tumour microenvironment can exhibit both pro- and anti-oncogenic properties. Therefore, in order to develop new cancer therapy methods using MSCs, it is necessary to further deepen the knowledge, leading to understanding of the molecular and cellular interactions between MSCs and the tumour niche; Abbreviations: MSCs: mesenchymal stromal cells; CSCs: cancer stem cells; BM-MSCs: bone-marrow-derived MSCs; A-MSCs: adipose tissue-derived MSCs; N-MSC: naïve MSCs; CAFs: cancer-associated fibroblasts; EVs: MSC-derived extracellular vesicles; CTLA-4: cytotoxic T-lymphocyte-associated protein 4; TCR: T-cell receptor; PD1/PD-1L: programmed death receptor 1/ programmed death receptor 1 ligand; MHC: major histocompatibility complex; TGF-β1: transforming growth factor beta 1; CXCL3L1: C-X-C motif chemokine ligand 3/ C-X-C motif chemokine ligand 3 like 1; TRAIL: TNF-related apoptosis-inducing ligand CCL5: C-C motif chemokine ligand 5; VEGF: Vascular endothelial growth factor; EGF: epidermal growth factor; PDGF: platelet-derived growth factor; VEGFR1: vascular endothelial growth factor receptor 1; NK4: HGF-antagonist/angiogenesis inhibitor; TK: herpes simplex virus (HSV)-1 thymidine kinase; CD: herpes simplex virus (HSV)-1 cytosine deaminase; PTEN: phosphatidylinositol 3,4,5-trisphosphate 3-phosphatase; CXCLs: chemokine (C-X-C motif) ligands; GTA: Ganciclovir; CDDP: cisplatin; DXR: Doxorubicin; PTX: Paclitaxel; GCB: Gemcitabine; 5-FU: 5-Fluorouracyl; CTLs: cytotoxic T cells; CAFs: cancer-associated fibroblasts; TAM2: tumour-associated macrophages M2; NK cells: activated natural killer cells; → activation mechanisms; ¦ inhibitory mechanisms; ↑: increase in expression and activity, ↓: decrease in expression and activity.

**Table 1 cells-13-01270-t001:** Interplay between MSCs in the tumour microenvironment and cancer cells in the selected in vitro models of HNC and OESCC.

Author	MSC in In Vitro Models of HNC and OESCC	
	Study Design	Mechanisms/Underlying Signalling Pathway/Results
Wats et al. [126]	- The tumour effect of BM-SSCs MSCs on JHU-011, JHU-012, and JHU-019 OSCC/OPSCC (oral cavity/oropharyngeal cancer) cell lines	- Tumour cells secretion of PDGF-AA and IL-6 → ↑ MSCs (CD90^+^, CD105^+^, GREM1^+^) recruitment, MSCs migration and invasion (*p* < 0.0001) and ↑ MSC chemotaxis → ↓ PDGFR-α (*p* < 0.0001), but not PDGFR-β
Kansy et al. [127]	- MSC were isolated from tumour tissues of HNSCC patients (Tu-MSC) - Tu-MSC expressed MSC surface markers (CD73, CD90 and CD105)	- Tu-MSC → ↑ inflammatory cytokines (IL-6, IL-8, TNF-α), homeostatic chemokines (SDF-1α) and T-helper type 1 cytokines (INF-γ) and ↑ TILs - HNSCC-derived factors → ↑ MSC activity and ↑ IL-8 CD54 (ICAM-1)
Ji et al. [128]	- MSCs from normal gingival tissue (GMSCs) were isolated and the effect of GMSCs on oral cancer cells (OCC) CAL-27 and WSU-HN6 were used; conditioned medium derived from GMSCs (GMSCs-CM) was studied	- GMSCs-CM → ↓ growth and ↑ cancer cells apoptosis (*p* < 0.05 and *p* < 0.05, respectively) - GMSCs → ↑ p-JNK, cleaved PARP, cleaved caspase-3, Bax and ↓ p-STAT3, p-ERK1/2, Bcl-2, CDK4, cyclin D1, PCNA and ↓ surviving molecules
Wu et al. [130]	- Human tongue cancer cell (TSCC) lines CAL-27 and TSCCA were cultured in BMSCs-conditioned media (MSC-CM)	- MCS-derived CCN2 → ↑ tumour cell proliferation, migration and invasion
Liu et al. [131]	- Human head and neck (HNC) CAL-27 and HN4 cell lines were cultured in BM-MSCs-conditioned media (MSC-CM)	- CAL-27 and HN4 cells cultured in MSC-CM → ↓ E-cadherin and P16/P21 mRNA (*p* < 0.05 and *p* < 0.01), and ↑ PI3K/Akt/mTOR, POSTN, Snail, Twist, N-cadherin mRNA (all *p* < 0.05) and ↑HNC migration
Li et al. [133]	- MSCs were derived from normal oral mucosa (N-MSC), oral leucoplakia with dysplasia (LK-MSC) and oral carcinoma and oral carcinoma in situ tissues (Ca-MSC)	- LK-MSC exhibited ↓ proliferation and migration and were more sensitive to the TGF-β1 stimulation, compared with the N-MSCs and Ca-MSCs - LK-MSC- and Ca-MSC-derived exosomes encapsulating miR-8485 → ↑ proliferation, migration and invasion abilities of epithelial cells
Shi et al. [136]	- The tumour effect of bone marrow-derived human mesenchymal stem cells (BM-MSCs) on nasopharyngeal carcinoma (NPC) CNE1, CNE2, 5-8F, 6-10B cell lines	- MSC-exosomes by FGF19 production → ↑ CNE2 cell invasion and metastasis - CNE2 cells treated with FGF19 → ↓ E-cadherin and ↑ N-cadherin and vimentin expression - FGFR4 siRNA reversed MSC-exosomes-induced ERK phosphorylation and EMT modulation
Wang et al. [137]	- The involvement of MSCs-derived cytokines on Eca109 and TE-1 human oesophageal cancer cell (OESCC) lines were examined	- MSCs-derived B2M → ↑ JAK/STAT pathway → ↑ tumour-initiation and invasion, and ↑ drug resistance in ESCC cells
Nakayama et al. [139]	- The co-culture of adipose tissue AT fragment (ATF)-embedded collagen gel and the human oesophageal squamous cell carcinoma (ESCC) cell lines, EC-GI-10 and TE-9	- Both EC-GI-10 and TE-9 cells → ↓ MSC-like cells from ATFs - ATFs → ↑ MAPK and PI3K-AKT pathways and ↓ IGF-1R and HER2 in ESCC
Wang et al. [140]	- Cell fusion of human umbilical cord mesenchymal stem cells (hMSCs) with human oesophageal cancer (EC) cell line EC9706	- hMSCs-ECs → ↓ cell growth and tumorigenicity, and ↑ tumour apoptosis and DUSP6/MKP3 in MAPK pathway
Li et al. [141]	- Human mesenchymal stem cells (hMSCs) derived from BM-MSCs of healthy donors were co-cultured with oesophageal cancer cell line (Eca-109)	- hMSCs → ↓ proliferation and invasion of Eca-109 cells as a result of WNT signalling pathway - hMSCs → ↑apoptosis of tumour cells, PCNA, Cyclin E, pRb, Bcl-2, Bcl-xL, and ↓ MMP-2
Liotta et al. [142]	- Bone marrow-derived human mesenchymal stem cells (BM-MSCs) were isolated fresh tissue samples form HNSCC patients - The co-culture of tumour-MSCs with anti-CD3 plus anti-CD28 or MLR-stimulated CD4^+^ or CD8^+^ T cells	- Tumour-MSCs showed a clear immunosuppressive activity on stimulated T lymphocytes - tumour-MSCs → ↓ proliferation and cytokine production of activated CD4^+^ and CD8^+^ T cells in a dose-dependent fashion through indoelamine 2,3 dioxygenase activity (IDO) activity and ↑ chemotactic activity of T cells
Mazzoini et al. [143]	- HNSCC cell lines were derived from tumoral specimens from seven patients - PBMCs were obtained from healthy donor - The HNSCC-MSC co-culture was examined	- HNSCC-MSCs treated or not with IFN-γ and TNF-α → ↑ IDO-1 and IL4I1 mRNA expression levels - The proliferation rate of CD4+ T cells was significantly reduced by the addition in culture of HNSCC-MSC in a dose-dependent manner
Rowan et al. [147]	- Human CAL-27 and SCC-4 head and neck cancer cells (HNSCC) cell lines were co-cultured human adipose tissue-derived stromal cells (ASCs)	- HNSCC-ASC-CM (condition medium derived from co-culture) had no effect on cell growth - HNSCC-ASC → ↑ migration of both CAL-27 and SCC-4 cells
Coccè et al. [148]	- MSCs from gingival papilla (GinPa-MSCs) were isolated; an anti-cancer activity of the drug-releasing GinPa-MSCs against a tongue squamous cell carcinoma cell line (SCC154) was studied	- GinPa-MSCs incorporated the drugs and then released them in active form → ↓ proliferation and growth of SCC154 - The highest sensitivity of SCC154 cell line to Gemcitabine (GCB) according to a dose-response kinetics was observed
Wang et al. [149]	- The tumour effect of bone marrow-derived human mesenchymal stem cells (BM-MSCs) on oral/oropharyngeal squamous carcinoma (OSCC) JHU-012, JHU-019 cell lines	- Co-culture of MSCs with OSCC cell lines → ↑ PDGF-AA, PDGFR-α Bcl-2 and MCP-1 and ↓ tumour apoptosis, Bid and AKT pathway - JHU-012 cells grown in co-culture with MSCs were more susceptible to cisplatin (CDDP) following pretreatment with, Crenolanib, a PDGFR inhibitor compared to cancer cells grown alone (*p* < 0.0001)
Liu et al. [150]	- Co-cultures of bone marrow-derived human mesenchymal stem cells (BM-MSCs) with HNSCC SCC-25 and HSC-2 cell lines were established; the role of BM-MSCs on tumour progression and chemotherapy resistance to Paclitaxel was estimated	- Co-culture of MSCs with SCC-25 cancer cells → ↑ POSTN, GDF11, IGFBP5, CXCL11, PI3K/PTEN/AKT and ↓ DAPK1 - Naïve SCC-25 cells were more sensitive to Paclitaxel, compared to the MSC/HNSCC fused cells; ↓ caspase 3 and cleaved-PARP-1 was observed
Salo et al. [151]	- The tumour effect of BM-MSCs on HSC-3 oral tongue squamous cell carcinoma (OTSCC) cell lines	- BM-MSCs products i.e., chemokine CCL5 → ↓ proliferation and ↑ invasion of OTSCC cells, and ↑ inflammatory chemokines and ↑ type I collagen mRNA in OTSCC cells; antibody against CCL5 inhibited cancer invasion
Hong et al. [152]	- The tumour effect of the conditioned medium from mesenchymal stromal cells (MSCs-CM) on human oesophageal squamous cell carcinoma (OESCC) ECa109 and TE-1 cell lines	- MSCs-CM → ↑ proliferation, viability and invasion of ESCC; GREM1 was overexpressed in ESCC - The TGF-β/BMP4 pathway participated in the conditioned medium from shGREM1-MSCs (shGREM1-MSCs-CM) induced anti-tumour effect on enhanced oesophageal malignancy induced by MSCs-CM treatment
Mazzoini et al. [153]	- HNSCC and normal tissue specimens were obtained during surgical procedure; PBMCs were also taken - Bone-marrow-derived (BM-MSC) and HNSCC-derived mesenchymal stromal cells (HNSCC-MSC) were maintained	- HNSCC-MSC → ↑ CD4^+^ and CD8^+^ T cells with a tissue-resident memory cells phenotype (Trm) in a VCAM1-dependent manner - HNSCC-MSC → ↑ IL-7, IL-15, PD-L1, and the Notch ligand DII1

HNSCC: head and neck squamous cell carcinoma; MSCs: mesenchymal stromal cells; BM-MSCs: bone marrow-derived mesenchymal stem cells; OSCC: oral squamous cell carcinoma; OPSSC: oropharyngeal squamous cell carcinoma; OTSCC/TSCC: oral tongue squamous cell carcinoma; OESCC: oesophageal squamous cell carcinoma; NPC: nasopharyngeal carcinoma; Ca-MSC/Tu-MSC: cancer stem cells/MSCs isolated form tumour tissues; GMSCs: MSCs from normal gingival tissue; ASCs: human adipose tissue-derived stromal cells; TILs: tumour infiltrating lymphocytes; EMT: epithelial–mesenchymal transition; PINP: type I collagen N-terminal propeptide; POSTN: periostin/osteoblast-specific factor-2; GREM-1: Gremlin-1; DOK: dysplastic oral keratinocytes; PI3K/Akt/mTOR: phosphoinositide 3-kinase/mammalian target of rapamycin; PDGF-AA: platelet-derived growth factor; PDGF-R: platelet-derived growth factor receptor; CAL-27 and HN4: human HNC cell lines; SDF-1α: stromal cell-derived factor-1α; TNF-α: tumour necrosis factor alpha; CD54/ ICAM-1: intercellular adhesion molecule 1; MMP-2/9: matrix metalloproteinases 2 and 9; JNK: c-Jun N-terminal kinases; PARP: poly (ADP-ribose) polymerase 1; Bax: apoptosis regulator, a member of the Bcl-2 gene family; Bid: BH3-interacting domain death agonist; STAT3: signal transducer and activator of transcription 3; ERK1/2: extracellular signal-regulated kinases 1 and 2; CCL5: C-C motif chemokine ligand 5; CDK4: cyclin-dependent kinase 4; PCNA: proliferating cell nuclear antigen; B2M: β2-microglobulin; CCN2/CTGF: cysteine-rich protein/connective tissue growth factor; GinPa-MSCs: MSCs isolated and expanded from gingival papilla; N-MSC: MSCs derived from normal oral mucosa; LK-MSC: MSCs derived from oral premalignant lesion; PCNA: proliferating cell nuclear antigen; pRb: phospho-retinoblastoma protein; Bcl-2: B-cell lymphoma/leukemia-2; CDK2: cyclin E-cyclin-dependent kinase 2; TGF-β/BMP: transforming growth factor-beta/bone morphogenetic protein; IDO: indoelamine 2,3 dioxygenase; Trm: tissue-resident memory cells phenotype; ↑: increase in expression and activity, ↓: decrease in expression and activity.

**Table 2 cells-13-01270-t002:** Interaction between MSCs in the tumour microenvironment and cancer cells in the selected animal/in vivo models of HNC and OESCC.

Author	MSCs in Animal and In Vivo Models of HNC and OESCC	
	Study Design	Mechanisms/Underlying Signalling Pathway/Results
Kansy et al. [127]	- The impact of tumour-derived MSC (Tu-MSC) on tumour growth in HNSCC xenograft murine model HNSCC cells (FaDu and UM-SSC-22B)	- Tu-MSC → ↑ stromal support for human HNSCC cell lines in vivo and ↑ cancer growth in a murine model
Ji et al. [128]	- Isolated MSCs from normal gingival tissue (GMSCs) were used - Animal model (BALB/C nude mice) after oral cancer cells (OCC) and GMSCs co-injection was analysed	- GMSCs → ↑ anti-cancer effect via altering EMT in nude mice - GMSCs → ↓ growth of OCC CAL-27 cells in vivo
Wu et al. [130]	- Human tongue squamous cell carcinoma (TSCC) samples FFPE tissue blocks of 90 patients were analysed - MSCs were co-injected with TSCC cells (TSCCA and CAL-27) into immune-deficient SCID mice	- CCN2 induced by MSCs → ↑ proliferation of TSCCA and CAL-27 cell lines in vivo
Liu et al. [131]	- Murine model of HNSCC/OSCC carcinogenesis (male BALB/C nude mice) was analysed - Mixed cells of CAL-27 and BM-MSC (MSC-CM) were injected into the tongue of the mice	- BM-MSCs → ↑ tumour growth, invasion, formation of metastatic lesions and ↑ POSTN in EMT - MSC-CM → ↑ N-cadherin and ↓ E-cadherin than in the control group (*p* < 0.05 and *p* < 0.01)
Shi et al. [135]	- The female NOD/SCID mice were subcutaneously inoculated with the nasopharyngeal carcinoma (NPC) CNE2 cells and intratumorally injected with MSC-exosomes or PBS as a control	- MSC-exosomes → ↑ tumour growth in nude mice - MSC-exosomes → ↑ N-cadherin and vimentin and ↓ E-cadherin-positive CNE2 tumour cells
Wang et al. [137]	- Murine xenograft model of oesophageal squamous cell carcinoma (OESCC) (male BALB/C nude mice) after subcutaneous injection of MSCs was analysed - Samples from 30 ESCC patients after resection who received first-line chemotherapy were used	- MSCs-derived B2M → ↑ tumour development in vivo - MSCs-derived B2M → poor prognosis (↓ PFS) of ESCC patients (*p* = 0.039)
Wang et al. [140]	- Xenograft immunodeficient female athymic nude mice (BALB/C nu/nu) and SCID mice were used - The oesophageal tumorigenesis in mice engraftments with human umbilical cord stem cell transplantation (hMSCs) was analysed	- Fusion of hMSCs with oesophageal carcinoma cells → ↓ tumorigenicity and cell growth (via DUSP6/MKP3 in MAPK pathway), and ↑ apoptosis
Tian et al. [141]	- BALB/C nude mice were injected subcutaneously with oesophageal cancer cell line (Eca-109 cells) and human mesenchymal stem cells (hMSCs)	- hMSCs → ↑ tumour formation and volume, vascularity and tumour growth
Liotta et al. [142]	- MSCs isolated from fresh tumoral tissue specimens of HNSCC patients and normal samples were obtained	- CD90^+^ tumour-MSC → ↑ tumour dimension (*p* < 0.001) and ↓ TILs frequency
Rowan et al. [147]	- Xenograft female NUDE mice (BALB/C) were used - Human CAL-27 and SCC-4 head and neck cancer cells (HNSCC) were co-cultured with adipose tissue-derived stromal/stem cells (ASCs)	- ASCs → ↑ MMP-2 and MMP-9, angiogenesis and early micrometastasis of CAL-27 tumour xenografts to mouse brain
Liu et al. [150]	- MSCs derived from human bone marrow with HNSCC (SCC-25 and HSC-2) cells were used - Naïve SCC-25, sorted SCC-25 or fused MSC/SCC cells were injected into the tongue of the xenograft male BALB/C nude mice	- Sorted SCC and MSC/SCC fused cells → ↑ aggressive tumour pattern, compared to the naïve SCC-25 cells - MSC-HNSCC cells → ↑ drug resistance to Paclitaxel through epigenetic modifications in mice grafted
Hong et al. [152]	- Human oesophageal squamous cell carcinoma (OESCC) ECa109, TE-1 cell lines were used - ESCC cells with or without the shRNA silencing of GREM1 in MSCs (shGREM1-MSCs) were subcutaneously injected into BALB/C nude mice	- MSCs-CM → ↑ malignancy of xenograft oesophageal tumours in vivo - shGREM1-MSCs reversed the malignancy and EMT of ESCC in vivo *(*↑ E-cadherin and ↓ β-catenin, vimentin and N-cadherin)
Chen et al. [154]	- A chemically-induced oral carcinogenesis model by 4-nitroquinoline-1-oxide (4NQO), that generated precancerous lesions and cancerous lesions in the oral cavity of rats was used	- MSCs were enriched in carcinogen-induced dysplasia and oral cancers - Lesion-derived MSCs → ↑ cellular proliferation and ↓ T-cell proliferation
Bruna et al. [156]	- Oral squamous cell carcinoma (OSCC) was induced in Syrian hamsters by topical application of dimethylbenz[a]anthracene (DMBA) in the buccal pouch - Local administration of allogenic bone marrow-derived MSCs (BM-MSCs) was performed	- MSCs administration → ↑ proliferation and genes expression i.e., ECGR2, BTC, TRIM2, EAF2 and ↓ tumour cell apoptosis - MSC injection modified neither the density of vasculature nor the degree of inflammation in OSCC tumours
Bruna et al. [157]	- Oral squamous cell carcinoma (OSCC) was induced in Syrian golden hamsters by topical application of 7,12-dimethylbenz[a]anthracene in buccal pouch - At the hyperplasia, dysplasia, or papilloma stage, animals received intracardially the allogeneic bone marrow-derived (BM-MSCs)	- BM-MSCs administered at papilloma stage did not progress to carcinoma stage - Animals receiving BM-MSCs at hyperplasia stage developed tumours larger than those found in control animals (*p* < 0.05)
Tan et al. [160]	- Athymic nude mice with FaDu human head and neck cancer xenografts were used - E1-MYC-derived exosomes (an MSC cell line immortalised with the MYC gene), were injected subcutaneously into athymic nude mice	- E1-MYC do not form tumours in nude mice - E1-MYC-derived exosomes did not inhibit or promote tumour growth
Zielske et al. [161]	- The effect of radiotherapy on the localisation of lentivirus-transduced MSCs to head and neck carcinoma xenografts (HNSCC-MSCC1 xenografts) was assessed	- Irradiation did not increased MSC localisation in HNSCC-MSCC1 xenografts

HNSCC: head and neck squamous cell carcinoma; MSCs: mesenchymal stromal cells; BMSSCs: bone marrow-derived mesenchymal stem cells; ASCs: adipose tissue-derived stromal/stem cells; OSCC: oral squamous cell carcinoma; OPSSC: oropharyngeal squamous cell carcinoma; OTSCC/TSCC: oral tongue squamous cell carcinoma; OESCC: oesophageal squamous cell carcinoma; NPC: nasopharyngeal carcinoma; Tu-MSC: MSCs isolated form tumour tissues; GMSCs: MSCs from normal gingival tissue; CAFs: cancer-associated fibroblasts; EMT: epithelial-mesenchymal transition; PFS: progression-free survival; FFPE: formalin-fixed, paraffin-embedded samples; TILs: tumour-infiltrating leukocytes; PINP: type I collagen N-terminal propeptide; POSTN: periostin/osteoblast-specific factor-2; GREM1: Gremlin-1; FaDu, CAL-27 and HN4: human HNC cell lines; DUSP6/MKP3: dual-specificity phosphatase specific for the MAP kinases ERK1/2; MMP-2/9: metalloproteinases 2 and 9; B2M: β2-microglobulin; CCN2/CTGF: cysteine rich protein/connective tissue growth factor; EGR2: early growth response 2; BTC: betacellulin; TRIM2: tripartite motif-containing protein 2; EAF2: ELL protein-associated factor 2; ↑: increase in expression and activity, ↓: decrease in expression and activity.

## Data Availability

Data are available in a publicly accessible repository.

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
