# Peer review of "Role of Mesenchymal Stem/Stromal Cells in Head and Neck Cancer—Regulatory Mechanisms of Tumorigenic and Immune Activity, Chemotherapy Resistance, and Therapeutic Benefits of Stromal Cell-Based Pharmacological Strategies"

_cells, 2024, doi:10.3390/cells13151270_

Round 1

Reviewer 1 Report

Comments and Suggestions for Authors

The manuscript titled "The Role of Mesenchymal Stem/Stromal Cells in Head and Neck Cancer – Regulatory Mechanisms of Tumorigenic and Immune Activity, Chemotherapy Resistance, and Therapeutic Benefits of the Use of Stromal Cell-Based Pharmacological Strategies" by Katarzyna Starska-Kowarska.

It will be improved if the followings are undertaken:

- The globe cases and rates of head and neck cancer should be elaborated more in the introduction, and the design of figures such as pie chart to show these numbers would enhance the knowledge transfer.

- The abstract is far too long and should be reduced to around 250 words.

- Figure 1 is too busy, it should be enlarged to A4, landscape size for better visualization.

- Table 1 should be simplified as it is a bit long and clumsy.

- The number of references is too many, which should be reduced.

- Overall, the article is suffered from a bad article structure, it should be seriously arranged in order to let people to comprehend.

- At the end, the conclusion should be simplified to a single short paragraph, now is too long (more than one page).

- Typos and unfriendly mode of English usage can be found.

Comments on the Quality of English Language

- Coherence of word and font style should be ensured.

- The manuscript must be proofread by an English native speaker.

Reviewer 2 Report

Comments and Suggestions for Authors

This is an exhaustive narrative review on mesenchymal stem cells in head and neck cancers. I regret to inform you that I am not able to recommend its publication in Cells in the current form. First, the review does not have a clear protocol and does not follow the classical PRISMA guidelines, putting it very behind today’s standards. The lack of a clear search strategy to rescue the relevant articles left behind important studies. Moreover, the author stated that one of the inclusion criteria is “articles written in English”, but later stated that “there was no restriction on language”. In fact, the description of the methods in the end of the introduction is very limited and unusual, far from ideal. A protocol with a well-defined approach, search strategy, eligibility criteria, study selection, data extraction and data synthesis, quality assessment, among others (PRISMA guideline will help here) is important. It is important to assess the possible presence of bias in the individual studies included in the analysis. Second, the organization and description of the main findings is confused by combining different types of tumors which belong to the heterogeneous group of the head and neck cancers. By the way, cancers of the esophagus and nose are not usually classified as head and neck cancers. Third, Tables 1 and 2 are long, exhaustive and should be put as supplementary files, whereas Figure 1 is filled with lots of information, and it should be splitted in two, making it more readable.

Round 2

Reviewer 1 Report

Comments and Suggestions for Authors

Most of my concerns are settled. Thank you.

Reviewer 2 Report

Comments and Suggestions for Authors

I congratulate the author for her best efforts to address the reviewer’s comments to improve the manuscript.